# Reinforcement Learning Gradients as Vitamin for Online Finetuning Decision Transformers

**Kai Yan**     **Alexander G. Schwing**     **Yu-Xiong Wang**
University of Illinois Urbana-Champaign
{kaiyan3, aschwing, yxw}@illinois.edu
https://github.com/KaiYan289/RL_as_Vitamin_for_Online_Decision_Transformers

## Abstract

Decision Transformers have recently emerged as a new and compelling paradigm for offline Reinforcement Learning (RL), completing a trajectory in an autoregressive way. While improvements have been made to overcome initial shortcomings, online finetuning of decision transformers has been surprisingly under-explored. The widely adopted state-of-the-art Online Decision Transformer (ODT) still struggles when pretrained with low-reward offline data. In this paper, we theoretically analyze the online-finetuning of the decision transformer, showing that the commonly used Return-To-Go (RTG) that's far from the expected return hampers the online fine-tuning process. This problem, however, is well-addressed by the value function and advantage of standard RL algorithms. As suggested by our analysis, in our experiments, we hence find that simply adding TD3 gradients to the finetuning process of ODT effectively improves the online finetuning performance of ODT, especially if ODT is pretrained with low-reward offline data. These findings provide new directions to further improve decision transformers.

## 1 Introduction

While Reinforcement Learning (RL) has achieved great success in recent years [55, 31], it is known to struggle with several shortcomings, including training instability when propagating a Temporal Difference (TD) error along long trajectories [14], low data efficiency when training from scratch [67], and limited benefits from more modern neural network architectures [12]. The latter point differs significantly from other parts of the machine learning community such as Computer Vision [17] and Natural Language Processing [11].

To address these issues, Decision Transformers (DTs) [14] have been proposed as an emerging paradigm for RL, introducing more modern transformer architectures into the literature rather than the still widely used Multi-Layer Perceptrons (MLPs). Instead of evaluating state and state-action pairs, a DT considers the whole trajectory as a sequence to complete, and trains on offline data in a supervised, auto-regressive way. Upon inception, DTs have been improved in various ways, mostly dealing with architecture changes [37], the token to predict other than return-to-go [22], addressing the problem of being overly optimistic [46], and the inability to stitch together trajectories [5]. Significant and encouraging improvements have been reported on those aspects.

However, one fundamental issue has been largely overlooked by the community: *offline-to-online RL using decision transformers*, i.e., finetuning of decision transformers with online interactions. Offline-to-online RL [72, 41] is a widely studied sub-field of RL, which combines offline RL learning from given, fixed trajectory data and online RL data from interactions with the environment. By first training on offline data and then finetuning, the agent can learn a policy with much greater data efficiency, while calibrating the out-of-distribution error from the offline dataset. Unsurprisingly, this sub-field has become popular in recent years.

38th Conference on Neural Information Processing Systems (NeurIPS 2024).

While there are numerous works in the offline-to-online RL sub-field [35, 28, 62], surprisingly few works have discussed the offline-to-online finetuning ability of decision transformers. While there is work that discusses finetuning of decision transformers predicting encoded future trajectory information [64], and work that finetunes pretrained decision transformers with PPO in multi-agent RL [38], the current widely adopted state-of-the-art is the Online Decision Transformer (ODT) [74]: the decision transformer training is continued on online data following the same supervised-learning paradigm as in offline RL. However, this method struggles with low-reward data, as well as with reaching expert-level performance due to suboptimal trajectories [41] (also see Sec. 4).

To address this issue and enhance online finetuning of decision transformers, we theoretically analyze the decision transformer based on recent results [7], showing that the commonly used conditioning on a high Return-To-Go (RTG) that's far from the expected return hampers results. To fix, we explore the possibility of using *tried-and-true RL gradients*. Testing on multiple environments, we find that simply combining TD3 [21] gradients with the original auto-regressive ODT training paradigm is surprisingly effective: it improves results of ODT, especially if ODT is pretrained with low-reward offline data.

Our contributions are summarized as follows:

1) We propose a simple yet effective method to boost the performance of online finetuning of decision transformers, especially if offline data is of medium-to-low quality;

2) We theoretically analyze the online decision transformer, explain its "policy update" mechanism when using the commonly applied high target RTG, and point out its struggle to work well with online finetuning;

3) We conduct experiments on multiple environments, and find that ODT aided by TD3 gradients (and sometimes even the TD3 gradient alone) are surprisingly effective for online finetuning of decision transformers.

## 2 Preliminaries

**Markov Decision Process.** A Markov Decision Process (MDP) is the basic framework of sequential decision-making. An MDP is characterized by five components: the state space $S$, the action space $A$, the transition function $p$, the reward $r$, and either the discount factor $\gamma$ or horizon $H$. MDPs involve an *agent* making decisions in discrete steps $t \in \{0, 1, 2, \dots\}$. On step $t$, the agent receives the current state $s_t \in S$, and samples an action $a_t \in A$ according to its *stochastic* policy $\pi(a_t|s_t) \in \Delta(A)$, where $\Delta(A)$ is the probability simplex over $A$, or its *deterministic* policy $\mu(s_t) \in A$. Executing the action yields a reward $r(s_t, a_t) \in \mathbb{R}$, and leads to the evolution of the MDP to a new state $s_{t+1}$, governed by the MDP's transition function $p(s_{t+1}|s_t, a_t)$. The goal of the agent is to maximize the total reward $\sum_t \gamma^t r(s_t, a_t)$, discounted by the discount factor $\gamma \in [0, 1]$ for infinite steps, or $\sum_{t=1}^{H} r(s_t, a_t)$ for finite steps. When the agent ends a complete run, it finishes an *episode*, and the state(-action) data collected during the run is referred to as a *trajectory* $\tau$.

**Offline and Online RL.** Based on the source of learning data, RL can be roughly categorized into offline and online RL. The former learns from a given finite dataset of state-action-reward trajectories, while the latter learns from trajectories collected online from the environment. The effort of combining the two is called *offline-to-online* RL, which first pre-trains a policy using offline data, and then continues to finetune the policy using online data with higher efficiency. Our work falls into the category of offline-to-online RL. We focus on improving the decision transformers, instead of Q-learning-based methods which are commonly used in offline-to-online RL.

**Decision Transformer (DT).** The decision transformer represents a new paradigm of offline RL, going beyond a TD-error framework. It views a trajectory $\tau$ as a sequence to be auto-regressively completed. The sequence interleaves three types of tokens: **returns-to-go (RTG, the target total return)**, states, and actions. At step $t$, the past sequence of context length $K$ is given as the input, i.e., the input is $(\text{RTG}_{t-K}, s_{t-k}, a_{t-k}, \dots, \text{RTG}_t, s_t)$, and an action is predicted by the auto-regressive model, which is usually implemented with a GPT-like architecture [11]. The model is trained via supervised learning, considering the past $K$ steps of the trajectory along with the current state and the current return-to-go as the feature, and the sequence of all actions $a$ in a segment as the labels. At evaluation time, a **desired return RTG$_{\text{eval}}$** is specified, since the **ground truth future return RTG$_{\text{real}}$** isn't known in advance.

**Online Decision Transformer (ODT).** ODT has two stages: offline pre-training which is identical to classic DT training, and online finetuning where trajectories are iteratively collected and the policy is updated via supervised learning. Specifically, the action $a_t$ at step $t$ during rollouts is computed by the deterministic policy $\mu^{\mathrm{DT}}(s_{t-T:t}, a_{t-T:t-1}, \mathrm{RTG}_{t-T:t}, T = T_{\mathrm{eval}}, \mathrm{RTG} = \mathrm{RTG}_{\mathrm{eval}})$,[1] or sampled from the stochastic policy $\pi^{\mathrm{DT}}(a_t | s_{t-T:t}, a_{t-T:t-1}, \mathrm{RTG}_{t-T:t}, T = T_{\mathrm{eval}}, \mathrm{RTG} = \mathrm{RTG}_{\mathrm{eval}})$. Here, $T$ is the context length (which is $T_{\mathrm{eval}}$ in evaluation), and $\mathrm{RTG}_{\mathrm{eval}} \in \mathbb{R}$ is the target return-to-go. The data buffer, initialized with offline data, is gradually replaced by online data during finetuning.

When updating the policy, the following loss (we use the deterministic policy as an example, and thus omit the entropy regularizer) is minimized:

$$\sum_{t=1}^{T_{\mathrm{train}}} \left\| \mu^{\mathrm{DT}}\left(s_{0:t}, a_{0:t-1}, \mathrm{RTG}_{0:t}, \mathrm{RTG} = \mathrm{RTG}_{\mathrm{real}}, T = t\right) - a_t \right\|_2^2. \tag{1}$$

Note, $T_{\mathrm{train}}$ is the training context length and $\mathrm{RTG}_{\mathrm{real}}$ is the real return-to-go. For better readability, we denote $\{s_{x+1}, s_{x+2}, \ldots, s_y\}$, $x, y \in \mathbb{N}$ as $s_{x:y}$ (i.e., left *exclusive* and right *inclusive*), and similarly $\{a_{x+1}, a_{x+2}, \ldots, a_y\}$ as $a_{x:y}$ and $\{\mathrm{RTG}_{x+1}, \ldots, \mathrm{RTG}_y\}$ as $\mathrm{RTG}_{x:y}$. Specially, index $x = y$ represents an empty sequence. For example, when $t = 1$, $a_{0:0}$ is an empty action sequence as the decision transformer is not conditioned on any past action.

One important observation: the decision transformer is inherently *off-policy* (the exact policy distribution varies with the sampled starting point, context length and return-to-go), which effectively guides our choice of RL gradients to off-policy algorithms (see Appendix C for more details).

**TD3.** Twin Delayed Deep Deterministic Policy Gradient (TD3) [21] is a state-of-the-art online off-policy RL algorithm that learns a *deterministic* policy $a = \mu^{\mathrm{RL}}(s)$. It is an improved version of an actor-critic (DDPG [32]) with three adjustments to improve its stability: 1) *Clipped double Q-learning*, which maintains two critics (estimators for expected return) $Q_{\phi_1}, Q_{\phi_2} : |S| \times |A| \to \mathbb{R}$ and uses the smaller of the two values (i.e., $\min(Q_{\phi_1}, Q_{\phi_2})$) to form the target for TD-error minimization. Such design prevents overestimation of the $Q$-value; 2) *Policy smoothing*, which adds noise when calculating the $Q$-value for the next action to effectively prevent overfitting; and 3) *Delayed update*, which updates $\mu^{\mathrm{RL}}$ less frequently than $Q_{\phi_1}, Q_{\phi_2}$ to benefit from a better $Q$-value landscape when updating the actor. TD3 also maintains a set of *target networks* storing old parameters of the actor and critics that are soft-updated with slow exponential moving average updates from the current, active network. In this paper, we adapt this algorithm to fit the decision transformer architecture so that it can be used as an auxiliary objective in an online finetuning process.

## 3 Method

This section is organized as follows: we will first provide intuition why RL gradients aid online finetuning of decision transformers (Sec. 3.1), and present our method of adding TD3 gradients (Sec. 3.2). To further justify our intuition, we provide a theoretical analysis on how ODT fails to improve during online finetuning when pre-trained with low-reward data (Sec. 3.3).

### 3.1 Why RL Gradients?

In order to understand why RL gradients aid online finetuning of decision transformers, let us consider an MDP which only has a single state $s_0$, one step, a one dimensional action $a \in [-1, 1]$ (i.e., a bandit with continuous action space) and a simple reward function $r(a) = (a + 1)^2$ if $a \le 0$ and $r(a) = 1 - 2a$ otherwise, as illustrated in Fig. 1. In this case, a trajectory can be represented effectively by a scalar, which is the action. If the offline dataset for pretraining is of low quality, i.e., all actions in the dataset are either close to $-1$ or $1$, then the decision transformer will obviously not generate trajectories with high RTG after offline training. As a consequence, during online finetuning, the new rollout trajectory is very likely to be uninformative about how to reach $\mathrm{RTG}_{\mathrm{eval}}$, since it is too far from $\mathrm{RTG}_{\mathrm{eval}}$. Worse still, it cannot improve locally either, which requires $\frac{\partial \mathrm{RTG}}{\partial a}$. However, the decision transformer *yields exactly the inverse*, i.e., $\frac{\partial a}{\partial \mathrm{RTG}}$. Since the transformer is not invertible (and even if the transformer is invertible, often the ground truth $\mathrm{RTG}(a)$ itself is not), we cannot

---

[1]ODT uses different RTGs for evaluation and online rollouts, but we refer to both as $\mathrm{RTG}_{\mathrm{eval}}$ as they are both expert-level expected returns.

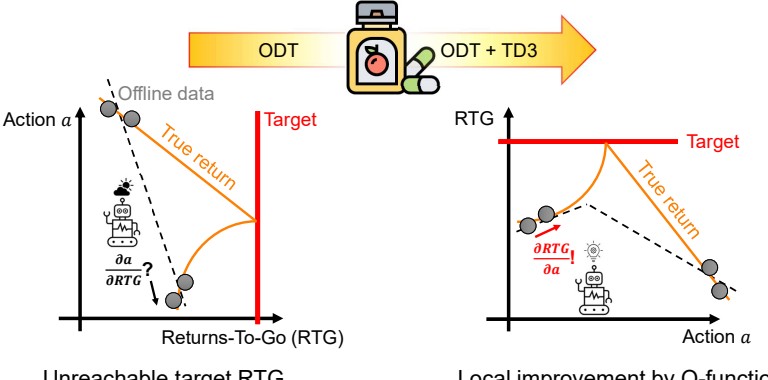

Unreachable target RTG          Local improvement by Q-function

Figure 1: An overview of our work, illustrating why ODT fails to improve with low-return offline data and RL gradients such as TD3 could help. The decision transformer yields gradient $\frac{\partial a}{\partial \text{RTG}}$, but local policy improvement requires the opposite, i.e., $\frac{\partial \text{RTG}}{\partial a}$. Therefore, the agent cannot recover if the current policy conditioning on high target RTG does not actually lead to high real RTG, which is very likely when the target RTG is too far from the pretrained policy and out-of-distribution. By adding a small coefficient for RL gradients, the agents can improve locally, which leads to better performance.

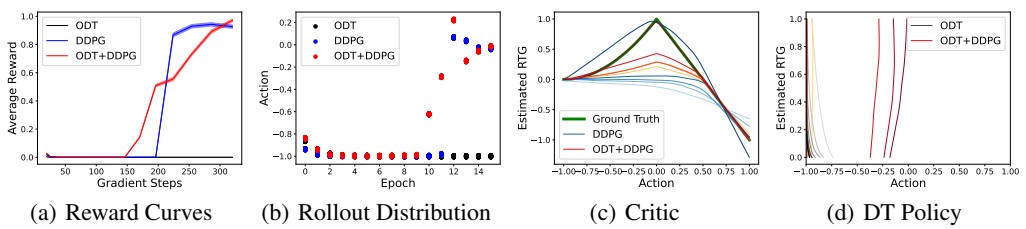

(a) Reward Curves          (b) Rollout Distribution          (c) Critic          (d) DT Policy

Figure 2: An illustration of a simple MDP, showing how RL can infer the direction for improvement, while online DT fails. Panels (a) and (b) show, DDPG and ODT+DDPG manage to maximize reward and find the correct optimal action quickly, while ODT fails to do so. Panel (c) shows how a DDPG/ODT+DDPG critic (from light blue/orange to dark blue/red) manages to fit ground truth reward (green curve). Panel (d) shows that the ODT policy (changing from light gray to dark) fails to discover the hidden reward peak near $0$ between two low-reward areas (near $-1$ and $1$ respectively) contained in the offline data. Meanwhile, ODT+DDPG succeeds in finding the reward peak.

easily estimate the former from the latter. Thus, the hope for policy improvement relies heavily on the generalization of RTG, i.e., policy yielded by high RTG$_{\text{eval}}$ indeed leads to better policy without any data as evidence, which is not the case with our constructed MDP and dataset.

In contrast, applying traditional RL for continuous action spaces to this setting, we either learn a value function $Q(s_0, a) : \mathbb{R} \to \mathbb{R}$, which effectively gives us a direction of action improvement $\frac{\partial Q(s_0, a)}{\partial a}$ (e.g., SAC [25], DDPG [32], TD3 [21]), or an advantage $A(s_0, a)$ that highlights whether focusing on action $a$ improves or worsens the policy (e.g., AWR [47], AWAC [40], IQL [28]). Either way provides a direction which suggests how to change the action locally in order to improve the (estimated) return. In our experiment illustrated in Fig. 2 (see Appendix F for details), we found that RL algorithms like DDPG [32] can easily solve the aforementioned MDP while ODT fails.

Thus, adding RL gradients aids the decision transformer to improve from given low RTG trajectories. While one may argue that the self-supervised training paradigm of ODT [74] can do the same by "prompting" the decision transformer to generate a high RTG trajectory, such paradigm is still unable to effectively improve the policy of the decision transformer pretrained on data with low RTGs. We provide a theoretical analysis for this in Sec. 3.3. In addition, we also explore the possibility of fixing this problem using other existing algorithms, such as JSRL [59] and slowly growing RTG (i.e., curriculum learning). However, we found that those algorithms cannot address this problem well. See Appendix G.8 for ablations.

## 3.2 Adding TD3 Gradients to ODT

In this work, we mainly consider TD3 [21] as the RL gradient for online finetuning. There are two reasons for selecting TD3. First, TD3 is a more robust off-policy RL algorithm compared to other off-policy RL algorithms [54]. Second, the success of TD3+BC [20] indicates that TD3 is a good candidate when combined with supervised learning. A more detailed discussion and empirical comparison to other RL algorithms can be found in Appendix C.

**Generally, we simply add a weighted standard TD3 actor loss to the decision transformer objective.** To do this, we follow classic TD3 and additionally train two critic networks $Q_{\phi_1}, Q_{\phi_2} : S \times A \to \mathbb{R}$ parameterized by $\phi_1, \phi_2$ respectively. In the offline pretraining stage, we use the following objective for the actor:

$$
\min_{\mu^{\mathrm{DT}}} \mathbb{E}_{\tau \sim D} \left[ \frac{1}{T_{\mathrm{train}}} \sum_{t=1}^{T_{\mathrm{train}}} \left[ -\alpha Q_{\phi_1}(s_t, \mu^{\mathrm{DT}}(s_{0:t}, a_{0:t-1}, \mathrm{RTG}_{0:t}, \mathrm{RTG} = \mathrm{RTG}_{\mathrm{real}}, T = t)) + \right. \right.
$$
$$
\left. \left. \| \mu^{\mathrm{DT}}(s_{0:t}, a_{0:t-1}, \mathrm{RTG}_{0:t}, \mathrm{RTG} = \mathrm{RTG}_{\mathrm{real}}, T = t) - a_t \|_2^2 \right] \right]. \tag{2}
$$

Here, $\alpha \in \{0, 0.1\}$ is a hyperparameter, and the loss sums over the trajectory segment. For critics $Q_{\phi_1}, Q_{\phi_2}$, we use the standard TD3 critic loss

$$
\min_{\phi_1, \phi_2} \mathbb{E}_{\tau \sim D} \sum_{t=1}^{T_{\mathrm{train}}} \left[ (Q_{\phi_1}(s_t, a_t) - Q_{\mathrm{min},t})^2 + (Q_{\phi_2}(s_t, a_t) - Q_{\mathrm{min},t})^2 \right], \quad \text{with}
$$
$$
Q_{\mathrm{min},t} = r_t + \gamma(1 - d_t) \min_{i \in \{1,2\}} Q_{\phi_{i,\mathrm{tar}}} \left( s_t, \mathrm{clip} \left( \mu_{\mathrm{tar}}^{\mathrm{RL}}(z_t) + \mathrm{clip}(\epsilon, -c, c), a_{\mathrm{low}}, a_{\mathrm{high}} \right) \right), \tag{3}
$$

where $\tau = \left\{ s_{0:T_{\mathrm{train}}+1}, a_{0:T_{\mathrm{train}}}, \mathrm{RTG}_{0:T_{\mathrm{train}}+1}, d_{0:T_{\mathrm{train}}}, r_{0:T_{\mathrm{train}}}, \mathrm{RTG} = \mathrm{RTG}_{\mathrm{real}} \right\}$ is the trajectory segment sampled from buffer $D$ that stores the offline dataset. Further, $d_t$ indicates whether the trajectory ends on the $t$-th step (true is 1, false is 0), $Q_{\mathrm{min}}$ is the target to fit, $Q_{\phi_{i,\mathrm{tar}}}$ is produced by the target network (stored old parameter), $z_t$ is the context for "next state" at step $t$. $\mu_{\mathrm{tar}}^{\mathrm{RL}}$ is the target network for the actor (i.e., decision transformer). For an $n$-dimensional action, $\mathrm{clip}(a, x, y), a \in \mathbb{R}^n, y \in \mathbb{R}^n, z \in \mathbb{R}^n$ means clip $a_i$ to $[y_i, z_i]$ for $i \in \{1, 2, \ldots, n\}$. $a_{\mathrm{low}} \in \mathbb{R}^n$ and $a_{\mathrm{high}} \in \mathbb{R}^n$ are the lower and upper bound for every dimension respectively.

To demonstrate the impact on aiding the exploration of a decision transformer, in this work we choose the simplest form of a critic, which is reflective, i.e., only depends on the current state. This essentially makes the $Q$-value an *average* of different context lengths sampled from a near-uniform distribution (see Appendix D for the detailed reason and distribution for this). The choice is based on the fact that training a transformer-based value function estimator is quite hard [45] due to increased input complexity (i.e., noise from the environment) which leads to reduced stability and slower convergence. In fact, to avoid this difficulty, many recent works on Large Language Models (LLMs) [13] and vision models [48] which finetune with RL adopt a policy-based algorithm instead of an actor-critic, despite a generally lower variance of the latter. In our experiments, we also found such a critic to be much more stable than a recurrent critic network (see Appendix G for ablations).

During online finetuning, we again use Eq. (2) and Eq. (3), but always use $\alpha = 0.1$ for Eq. (2).

While the training paradigm resembles that of TD3+BC, our proposed method improves upon TD3+BC in the following two ways: 1) **Architecture**. While TD3+BC uses MLP networks for single steps, we leverage a decision transformer, which is more expressive and can take more context into account when making decisions. 2) **Selected instead of indiscriminated behavior cloning.** Behavior cloning mimics all data collected without regard to their reward, while the supervised learning process of a decision transformer prioritizes trajectories with higher return by conditioning action generation on higher RTG. See Appendix G.9 for an ablation.

### 3.3 Why Does ODT Fail to Improve the Policy?

As mentioned in Sec. 3.1, it is the goal of ODT to "prompt" a policy with a high RTG, i.e., to improve a policy by conditioning on a high RTG during online rollout. However, beyond the intuition provided

in Sec. 3.1, in this section, we will analyze more formally why such a paradigm is unable to improve the policy given offline data filled with low-RTG trajectories.

Our analysis is based on the performance bound proved by Brandfonbrener et al. [7]. Given a dataset drawn from an underlying policy $\beta$ and given its RTG distribution $P_\beta$ (either continuous or discrete), under assumptions (see Appendix E), we have the following *tight* performance bound for a decision transformer with policy $\pi^{\text{DT}}(a|s,\text{RTG}_{\text{eval}})$ conditioned on $\text{RTG}_{\text{eval}}$:

$$\text{RTG}_{\text{eval}} - \mathbb{E}_{\tau=(s_1,a_1,...,s_H,a_H)\sim\pi^{\text{DT}}(a|s,\text{RTG}_{\text{eval}})}[\text{RTG}_{\text{real}}] \leq \epsilon \left( \frac{1}{\alpha_f} + 2 \right) H^2. \tag{4}$$

Here, $\alpha_f = \inf_{s_1} P_\beta(\text{RTG}_{\text{real}} = \text{RTG}_{\text{eval}}|s_1)$ for every initial state $s_1$, $\epsilon > 0$ is a constant, $H$ is the horizon of the MDP.[2] Based on this tight performance bound, we will show that **with high probability, $\frac{1}{\alpha_f}$ grows *superlinearly* with respect to RTG$_{\text{eval}}$**. If true, then the RTG$_{\text{real}}$ term (i.e., the actual return from online rollouts) must decrease to fit into the tight bound, as RTG$_{\text{eval}}$ grows.

To show this, we take a two-step approach: First, we prove that the probability mass of the RTG distribution is concentrated around low RTGs, i.e., *event probability* $\Pr_\beta (\text{RTG} - \mathbb{E}_\beta(\text{RTG}|s) \geq c|s)$ for $c > 0$ decreases superlinearly with respect to $c$. For this, we apply the Chebyshev inequality, which yields a bound of $O\left(\frac{1}{c^2}\right)$. However, without knowledge on $P_\beta(\text{RTG}|s)$, the variance can be made arbitrarily large by high RTG outliers, hence making the bound meaningless.

Fortunately, we have knowledge about the RTG distribution $P_\beta(\text{RTG}|s)$ from the collected data. If we refer to the maximum RTG in the dataset via $\text{RTG}_{\beta\text{max}}$ and if we assume all rewards are non-negative, then all trajectory samples have an RTG in $[0, \text{RTG}_{\beta\text{max}}]$. Thus, with adequate prior distribution, we can state that with high probability $1 - \delta$, the probability mass is concentrated in the low RTG area. Based on this, we can prove the following lemma:

**Lemma 1.** *(Informal) Assume rewards $r(s,a)$ are bounded in $[0, R_{max}]$,[3] and $RTG_{eval} \geq RTG_{\beta max}$. Then with probability at least $1 - \delta$, we have the probability of event $\Pr_\beta$ bounded as follows:*

$$\Pr_\beta \left( RTG_{eval} - V^\beta(s) \geq c|s \right) \leq O\left( \frac{R_{max}^2 T^2}{c^2} \right), \tag{5}$$

*where $\delta$ depends on the number of trajectories in the dataset and prior distribution (see Appendix E for a concrete example and a more accurate bound). $V^\beta(s)$ is the value function of the underlying policy $\beta(a|s)$ that generates the dataset, for which we have $V^\beta(s) = \mathbb{E}_\beta(RTG|s)$.*

The second step uses the bound of probability mass $\Pr_\beta(\text{RTG} \geq c|s)$ to derive the bound for $\alpha_f$. For the discrete case where the possibly obtained RTGs are finite or countably infinite (note, state and action space can still be continuous), this is simple, as we have

$$P_\beta \left( \text{RTG} = V^\beta(s) + c|s \right) = \Pr_\beta \left( \text{RTG} = V^\beta(s) + c|s \right) \leq \Pr_\beta \left( \text{RTG} \geq V^\beta(s) + c|s \right). \tag{6}$$

Thus $\alpha_f = \inf_{s_1} P_\beta(\text{RTG}|s_1)$ can be conveniently bounded by Lemma 1. For the continuous case, the proof is more involved as probability density $P_\beta(\text{RTG}|s)$ can be very high on an extremely short interval of RTG, making the total probability mass arbitrarily small. However, assuming that $P_\beta(\text{RTG}|s)$ is Lipschitz when $\text{RTG} \geq \text{RTG}_{\beta\text{max}}$ (i.e., RTG area not covered by dataset), combined with the discrete distribution case, we can still get the following (see Appendix E for proof):

**Corollary 1.** *(Informal) If the RTG distribution is discrete (i.e., number of possible different RTGs are at most countably infinite), then with probability at least- $1 - \delta$, $\frac{1}{\alpha_f}$ grows on the order of $\Omega(RTG_{eval}^2)$ with respect to $RTG_{eval}$. For continuous RTG distributions satisfying a Lipschitz continuous RTG density $p_\beta$, $\frac{1}{\alpha_f}$ grows on the order of $\Omega(RTG_{eval}^{1.5})$.*

Here, $\Omega(\cdot)$ refers to the big-Omega notation (asymptotic lower bound).

## 4 Experiments

In this section, we aim to address the following questions: **a)** Does our proposed solution for decision transformers indeed improve its ability to cope with low-reward pretraining data. **b)** Is improving

---

[2]Eq. (4) is very informal. See Appendix E for a more rigorous description.

[3]Note we use "max" instead of "$\beta$max" as this is a property of the environment and not the dataset.

what to predict, while still using supervised learning, the correct way to improve the finetuning ability of decision transformers? **c)** Does the transformer architecture, combined with RL gradients, work better than TD3+BC? **d)** Is it better to combine the use of RL and supervised learning, or better to simply abandon the supervised loss in online finetuning? **e)** How does online decision transformer with TD3 gradient perform compared to other offline RL algorithms? **f)** How much does TD3 improve over DDPG which was used in Fig. 2?

**Baselines.** In this section, we mainly compare to six baselines: the widely recognized state-of-the-art DT for online finetuning, Online Decision Transformer (**ODT**) [74]; **PDT**, a baseline improving over ODT by predicting future trajectory information instead of return-to-go; **TD3+BC** [20], a MLP offline RL baseline; **TD3**, an ablated version of our proposed solution where we use TD3 gradients only for decision transformer finetuning (but only use supervised learning of the actor for offline pretraining); **IQL** [28], one of the most popular offline RL algorithms that can be used for online finetuning; **DDPG [32]+ODT**, which is the same as our approach but with DDPG instead of TD3 gradients (for ablations using SAC [25], IQL [28], PPO [52], AWAC [40] and AWR [47], see Appendix C). Each of the baselines corresponds to one of the questions a), b), c), d), e) and f) above.

**Metrics.** We use the normalized average reward (same as D4RL's standard [19]) as the metric, where higher reward indicates better performance. If the final performance is similar, the algorithm with fewer online examples collected to reach that level of performance is better. We report the reward curve, which shows the change of the normalized reward's mean and standard deviation with 5 different seeds, with respect to the number of online examples collected. The maximum number of steps collected is capped at 500K (for mujoco) or 1M (for other environments). We also report evaluation results using the rliable [3] library in Fig. 7 of Appendix B.

**Experimental Setup.** We use the same architecture and hyperparameters such as learning rate (see Appendix F.2 for details) as ODT [74]. The architecture is a transformer with 4 layers and 4 heads in each layer. This translates to around 13M parameters in total. For the critic, we use Multi-Layer Perceptrons (MLPs) with width 256 and two hidden layers and ReLU [1] activation function. Specially, for the random dataset, we collect trajectories until the total number of steps exceeds 1000 in every epoch, which differs from ODT, where only 1 trajectory per epoch is collected. This is because many random environments, such as hopper, have very short episodes when the agent does not perform well, which could lead to overfitting if only a single trajectory is collected per epoch. For fairness, we use this modified rollout for ODT in our experiments as well. Not doing so does not affect ODT results since it does generally not work well on random datasets, but will significantly increase the time to reach a certain number of online transitions. After rollout, we train the actor for 300 gradient steps and the critic for 600 steps following TD3's delayed update trick.

### 4.1 Adroit Environments

**Environment and Dataset Setup.** We test on four difficult robotic manipulation tasks [49], which are the Pen, Hammer, Door and Relocate environment. For each environment, we test three different datasets: expert, cloned and human, which are generated by a finetuned RL policy, an imitation learning policy and human demonstration respectively. See Appendix F.1 for details.

**Results.** Fig. 3 shows the performance of each method on Adroit before and after online finetuning. TD3+BC fails on almost all tasks and often diverges with extremely large $Q$-value during online finetuning. ODT and PDT perform better but still fall short of the proposed method, TD3+ODT. Note, IQL, TD3 and TD3+ODT all perform decently well (with similar average reward as shown in Tab. 2 in Appendix B). However, we found that TD3 often fails during online finetuning, probably because the environments are complicated and TD3 struggles to recover from a poor policy generated during online exploration (i.e., it has a *catastrophic forgetting* issue). To see whether there is a simple fix, in Appendix G.7, we ablate whether an action regularizer pushing towards a pretrain policy similar to TD3+BC helps, but find it to hinder performance increase in other environments. IQL is overall much more stable than TD3, but improves much less during online finetuning than TD3+ODT. ODT can achieve good performance when pretrained on expert data, but struggles with datasets of lower quality, which validates our motivation. DDPG+ODT starts out well in the online finetuning stage but fails quickly, probably because DDPG is less stable compared to TD3.

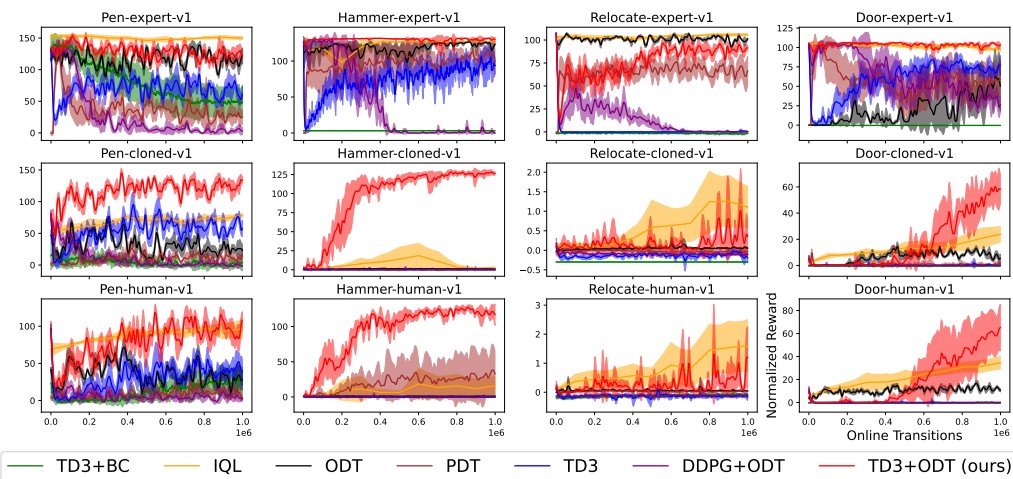

Figure 3: Results on Adroit [49] environments. The proposed method, TD3+ODT, improves upon baselines. Note that TD3, IQL, and TD3+ODT all perform decently at the beginning of online finetuning, but TD3 fails while TD3+ODT improves much more than IQL during online finetuning.

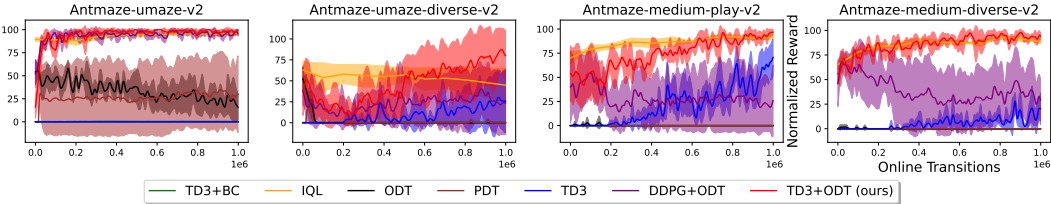

Figure 4: Reward curves for each method in Antmaze environments. IQL works best on the large maze, while our proposed method works the best on the medium maze and umaze. DDPG+ODT works worse than our method and IQL but much better than the rest of the baselines, which again validates our motivation that adding RL gradients to ODT is helpful.

## 4.2 Antmaze Environments

**Environment and Dataset Setup.** We further test on a harder version of the Maze2D environment in D4RL [19] where the pointmass is substituted by a robotic ant. We study six different variants, which are umaze, umaze-diverse, medium-play, medium-diverse, large-play and large-diverse.

**Results.** Fig. 4 lists the results of each method on umaze and medium maze before and after online finetuning (see Appendix C for reward curves and Appendix B for results summary on large antmaze). TD3+ODT works the best on umaze and medium maze, and significantly outperforms TD3. This shows that RL gradients alone are not enough for offline-to-online RL of the decision transformer. Though TD3+ODT does not work on large maze, we found that IQL+ODT works decently well. However, we choose TD3+ODT in this work because IQL+ODT does not work well on the random datasets. This is probably because IQL aims to address the Out-Of-Distribution (OOD) estimation problem [28], which makes it better at utilizing offline data but worse at online exploration. See Appendix C for a detailed discussion and results. DDPG+ODT works worse than TD3+ODT but much better than baselines except IQL.

## 4.3 MuJoCo Environments

**Environment and Dataset Setup.** We further test on four widely recognized standard environments [58], which are the Hopper, Halfcheetah, Walker2d and Ant environment. For each environment, we study three different datasets: medium, medium-replay, and random. The first and second one contain trajectories of decent quality, while the last one is generated with a random agent.

| | TD3+BC | IQL | ODT | PDT | TD3 | DDPG+ODT | TD3+ODT (ours) |
|---|---|---|---|---|---|---|---|
| Ho-M-v2 | 60.24(+4.4) | 44.72(-21.3) | **97.84(+48.69)** | 74.43(+72.21) | 88.98(+29.25) | 41.7(-13.18) | 89.07(+25.97) |
| Ho-MR-v2 | **99.07(+33.33)** | 62.76(-7.63) | 83.29(+63.17) | 84.53(+82.23) | 93.72(+55.66) | 32.36(+9.9) | 95.65(+65.89) |
| Ho-R-v2 | 8.36(-0.35) | 20.42(+12.36) | 29.08(+26.92) | 35.9(+34.67) | 75.68(+73.69) | 25.12(+23.14) | **76.13(+74.15)** |
| Ha-M-v2 | 51.29(+2.73) | 37.12(-10.35) | 42.27(+19.23) | 39.35(+39.55) | 70.9(+29.59) | 55.69(+14.71) | **76.91(+35.3)** |
| Ha-MR-v2 | 56.5(+13.07) | 49.97(+6.84) | 41.45(+26.77) | 31.47(+31.8) | 69.87(+40.59) | 53.71(+24.91) | **73.27(+43.98)** |
| Ha-R-v2 | 44.78(+31.12) | 47.85(+40.3) | 2.15(-0.09) | 0.74(+0.9) | **68.55(+66.3)** | 34.56(+32.31) | 59.35(+57.1) |
| Wa-M-v2 | 85.34(+3.49) | 65.55(-15.12) | 75.57(+18.47) | 63.37(+63.3) | 90.49(+24.74) | 2.01(-69.54) | **97.86(+27.08)** |
| Wa-MR-v2 | 83.28(+0.0) | 95.99(+28.78) | 77.2(+12.46) | 54.49(+54.18) | **100.88(+32.54)** | 1.04(-60.59) | 100.6(+42.54) |
| Wa-R-v2 | 6.99(+5.86) | 10.67(+4.96) | 14.12(+9.82) | 15.47(+15.32) | 69.91(+66.31) | 2.91(-2.47) | 57.86(+53.27) |
| An-M-v2 | 129.11(+7.11) | 110.36(+14.26) | 88.1(-0.51) | 52.08(+48.47) | 125.67(+37.55) | 10.81(-75.52) | **132.0(+41.42)** |
| An-MR-v2 | 129.33(+41.03) | 113.16(+24.24) | 85.64(+4.49) | 36.92(+32.41) | **133.58(+51.17)** | 4.05(-87.7) | 130.23(+52.08) |
| An-R-v2 | 67.89(+33.47) | 12.28(+0.97) | 24.96(-6.44) | 14.88(+10.38) | 63.47(+32.02) | 4.93(-26.55) | **71.69(+40.31)** |
| Average | 68.52(+14.6) | 55.9(+7.8) | 55.14(+18.58) | 41.97(+40.44) | 87.64(+44.95) | 22.87(-19.22) | **88.38(+46.59)** |

Table 1: Average reward for each method in MuJoCo environments before and after online finetuning. The best performance for each environment is highlighted in bold font, and any result $> 90\%$ of the best performance is underlined. To save space, the name of the environments and datasets are abbreviated as follows: for the environments Ho=Hopper, Ha=HalfCheetah, Wa=Walker2d, An=Ant; for the datasets M=Medium, MR=Medium-Replay, R=Random. The format is "final(+increase after finetuning)". The proposed solution performs well.

**Results.** Fig. 6 shows the results of each method on MuJoCo before and after online finetuning. We observe that autoregressive-based algorithms, such as ODT and PDT, fail to improve the policy on MuJoCo environments, especially from low-reward pretraining with random datasets. With RL gradients, TD3+BC and IQL can improve the policy during online finetuning, but less than a decision transformer (TD3 and TD3+ODT). In particular, we found IQL to struggle on most random datasets, which are well-solved by decision transformers with TD3 gradients. TD3+ODT still outperforms TD3 with an average final reward of $88.51$ vs. $84.23$. See Fig. 6 in Appendix B for reward curves.

**Ablations on $\alpha$.** Fig. 5 (a) shows the result of using different $\alpha$ (i.e., RL coefficients) on different environments. We observe an increase of $\alpha$ to improve the online finetuning process. However, if $\alpha$ is too large, the algorithm may get unstable.

**Ablations on evaluation context length $T_{\text{eval}}$.** Fig. 5 (b) shows the result of using different $T_{\text{eval}}$ on halfcheetah-medium-replay-v2 and hammer-cloned-v1. The result shows that $T_{\text{eval}}$ needs to be balanced between more information for decision-making and potential training instability due to a longer context length. As shown in the halfcheetah-medium-replay-v2 result, $T_{\text{eval}}$ too long or too short can both lead to performance drops. More ablations are available in Appendix G.

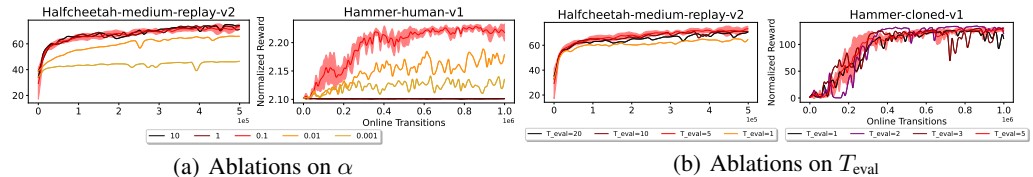

(a) Ablations on $\alpha$      (b) Ablations on $T_{\text{eval}}$

Figure 5: Panel (a) shows ablations on RL coefficient $\alpha$. While higher $\alpha$ aids exploration as shown in the halfcheetah-medium-replay-v2 case, it may sometimes introduce instability, which is shown in the hammer-human-v1 case. Panel (b) shows ablations on $T_{\text{eval}}$. $T_{\text{eval}}$ balances training stability and more information for decision-making.

## 5 Related Work

**Online Finetuning of Decision Transformers.** While there are many works on generalizing decision transformers (e.g., predicting waypoints [5], goal, or encoded future information instead of return-to-go [22, 5, 57, 36]), improving the architecture [37, 16, 53, 65] or addressing the overly-optimistic [46] or trajectory stitching issue [63]), there is surprisingly little work beyond online decision transformers that deals with online finetuning of decision transformers. There is some loosely related literature: MADT [31] proposes to finetune pretrained decision transformers with PPO. PDT [64] also studies online finetuning with the same training paradigm as ODT [74]. QDT [66] uses an offline RL

algorithm to re-label returns-to-go for offline datasets. AFDT [76] and STG [75] use decision transformers offline to generate an auxiliary reward and aid the training of online RL algorithms. A few works study in-context learning [33, 34] and meta-learning [60, 30] of decision transformers, where improvements with evaluations on new tasks are made possible. However, none of the papers above focuses on addressing the general online finetuning issue of the decision transformer.

**Transformers as Backbone for RL.** Having witnessed the impressive success of transformers in Computer Vision (CV) [17] and Natural Language Processing (NLP) [11], numerous works also studied the impact of transformers in RL either as a model for the agent [45, 38] or as a world model [39, 50]. However, a large portion of state-of-the-art work in RL is still based on simple Multi-Layer Perceptrons (MLPs) [35, 28]. This is largely because transformers are significantly harder to train and require extra effort [45], making their ability to better memorize long trajectories [42] harder to realize compared to MLPs. Further, there are works on using transformers as feature extractors for a trajectory [37, 45] and works that leverage the common sense of transformer-based Large Language Model's for RL priors [10, 9, 70]. In contrast, our work focuses on improving the new "RL via Supervised learning" (RvS) [7, 18] paradigm, aiming to merge this paradigm with the benefits of classic RL training.

**Offline-to-Online RL.** Offline-to-online RL bridges the gap between offline RL, which heavily depends on the quality of existing data while struggling with out-of-distribution policies, and online RL, which requires many interactions and is of low data efficiency. Mainstream offline-to-online RL methods include teacher-student [51, 6, 59, 72] and out-of-distribution handling (regularization [21, 29, 62], avoidance [28, 23], ensembles [2, 15, 24]). There are also works on pessimistic Q-value initialization [69], confidence bounds [26], and a mixture of offline and online training [56, 73]. However, all the aforementioned works are based on Q-learning and don't consider decision transformers.

## 6 Conclusion

In this paper, we point out an under-explored problem in the Decision Transformer (DT) community, i.e., online finetuning. To address online finetuning with a decision transformer, we examine the current state-of-the-art, online decision transformer, and point out an issue with low-reward, sub-optimal pretraining. To address the issue, we propose to mix TD3 gradients with decision transformer training. This combination permits to achieve better results in multiple testbeds. Our work is a complement to the current DT literature, and calls out a new aspect of improving decision transformers.

**Limitations and Future Works.** While our work theoretically analyzes an ODT issue, the conclusion relies on several assumptions which we expect to remove in future work. Empirically, in this work we propose a simple solution orthogonal to existing efforts like architecture improvements and predicting future information rather than return-to-go. To explore other ideas that could further improve online finetuning of decision transformers, next steps include the study of other environments and other ways to incorporate RL gradients into decision transformers. Other possible avenues for future research include testing our solution on image-based environments, and decreasing the additional computational cost compared to ODT (an analysis for the current time cost is provided in Appendix H).

## Acknowledgments

This work was supported in part by NSF under Grants 2008387, 2045586, 2106825, MRI 1725729, NIFA Award 2020-67021-32799, the IBM-Illinois Discovery Accelerator Institute, the Toyota Research Institute, and the Jump ARCHES endowment through the Health Care Engineering Systems Center at Illinois and the OSF Foundation.

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

# Appendix: Reinforcement Learning Gradients as Vitamin for Online Finetuning Decision Transformers

The Appendix is organized as follows. In Sec. A, we discuss the potential positive and negative social impact of the paper. Then, we summarize the performance shown in the main paper in Sec. B. After this, we will explain our choice of RL gradients in the paper in Sec. C, and why our critic serves as an average of policies generated by different context lengths in Sec. D. We then provide rigorous statements for the theroetical analysis appearing in the paper in Sec. E, and list the environment details and hyperparameters in Sec. F. We then present more experiment and ablation results in Sec. G. Finally, we list our computational resource usage and licenses of related assets in Sec. H and Sec. I respectively.

## A   Broader Societal Impacts

Our work generally helps automation of decision-making by improving the use of online interaction data of a pretrained decision transformer agent. While this effort improves the efficiency of decision-makers and has the potential to boost a variety of real-life applications such as robotics and resource allocation, it may also cause several negative social impacts, such as potential job losses, human de-skilling (making humans less capable of making decisions without AI), and misuse of technology (e.g., military).

## B   Performance Summary

In this section, we summarize the average reward achieve by each method on different environments and datasets, where the result for Adroit is shown in Tab. 2, and the result for Antmaze is shown in Tab. 3. As the summary table for MuJoCo is already presented in Sec. 4, we show the reward curves in Fig. 6. For a more rigorous evaluation, we also report other metrics including the median, InterQuartile Mean (IQM) and optimality gap using the rliable [3] library. See Fig. 7 for details. Breakdown analysis for each environment can be downloaded by browsing to `https://kaiyan289. github.io/assets/breakdown_rliable.rar`.

| | TD3+BC | IQL | ODT | PDT | TD3 | DDPG+ODT | TD3+ODT (ours) |
|---|---|---|---|---|---|---|---|
| P-E-v1 | 47.88(-84.23) | **149.65(-3.63)** | 121.82(+5.48) | 25.07(+25.56) | 61.56(-69.74) | 2.75(-129.63) | 120.65(-11.91) |
| P-C-v1 | 3.75(-10.8) | 78.12(+25.01) | 22.88(-24.0) | 14.05(+12.15) | 58.04(-17.39) | -1.41(-81.87) | **133.77(+58.05)** |
| P-H-v1 | 26.77(+3.19) | 96.5(+27.79) | 27.55(-13.91) | 4.03(+0.38) | 38.58(-57.71) | 2.09(-92.56) | **107.1(+11.87)** |
| H-E-v1 | 3.11(-0.02) | 126.54(+13.28) | 123.07(+12.79) | 98.95(+98.94) | 93.99(-31.41) | -0.24(-127.05) | **129.8(+6.34)** |
| H-C-v1 | 0.33(+0.03) | 2.27(+0.58) | 0.84(+0.32) | 0.66(+0.66) | 0.07(-0.69) | 0.12(-0.83) | **126.39(+124.59)** |
| H-H-v1 | 0.17(-0.3) | 16.12(+14.18) | 0.97(-0.13) | 32.76(+32.75) | -0.03(-1.11) | -0.06(-1.02) | **116.83(+115.82)** |
| D-E-v1 | -0.34(-0.01) | 97.57(-7.92) | 50.26(+50.14) | 59.48(+59.41) | 76.92(-25.56) | 26.48(-78.72) | **103.13(-1.94)** |
| D-C-v1 | -0.36(-0.01) | 23.8(+21.66) | 5.45(+5.37) | 1.38(+1.54) | 0.17(-4.8) | -0.01(-4.45) | **58.28(+53.31)** |
| D-H-v1 | -0.33(-0.1) | 34.64(+29.65) | 10.61(+6.69) | 0.05(+0.22) | -0.14(-9.22) | 12.39(-12.33) | **65.24(+55.94)** |
| R-E-v1 | -1.37(+0.22) | **105.78(+2.81)** | 101.16(+2.11) | 66.57(+66.7) | 0.44(-106.67) | 0.26(-106.48) | 91.38(-16.19) |
| R-C-v1 | -0.3(+0.0) | **1.1(+0.97)** | 0.06(+0.08) | -0.03(+0.04) | -0.19(-0.29) | -0.12(-0.32) | 0.36(+0.26) |
| R-H-v1 | -0.08(+0.1) | **1.6(+1.5)** | 0.04(+0.05) | 0.04(+0.17) | -0.17(-0.28) | -0.1(-0.27) | 1.19(+0.99) |
| Average | 6.6(-7.66) | 61.14(+10.49) | 38.73(+3.75) | 25.25(+24.87) | 31.85(-29.63) | 3.51(-52.96) | **87.84(+33.09)** |

Table 2: Average reward for each method in Adroit Environments before and after online finetuning. The best result for each setting is marked in bold font and all results $> 90\%$ of the best performance are underlined. To save space, the name of the environments and datasets are abbreviated as follows: P=Pen, H=Hammer, D=Door, R=Relocate for environment, and E=Expert, C=cloned, H=Human for the dataset. It is apparent that while both IQL, TD3 and TD3+ODT perform decently well before online finetuning, our proposed solution significantly outperforms all baselines on the adroit testbed. DDPG+ODT starts out well in the online stage, but fails probably due to DDPG's training instability compared to TD3.

## C   Why Do We Choose TD3 to Provide RL Gradients?

In this section, we provide an ablation analysis on which RL gradient fits the decision transformer architecture best. Fig. 8 illustrates the result of using a pure RL gradient for online finetuning of a pretrained decision transformer (for those RL algorithms with stochastic policy, we adopt the same

|        | TD3+BC | IQL | ODT | PDT | TD3 | DDPG+ODT | TD3+ODT (ours) |
|--------|--------|-----|-----|-----|-----|----------|----------------|
| U-v2   | 0.21(+0.07) | 95.99(+6.59) | 15.92(-38.08) | 29.94(+29.94) | 0.0(+0.0) | 95.62(+43.62) | **99.59(+83.59)** |
| UD-v2  | 0.53(+0.33) | 44.83(-18.17) | 0.0(-52.0) | 0.0(+0.0) | 25.55(+25.55) | 23.41(-14.59) | **80.0(+42.0)** |
| MP-v2  | 0.0(+0.0) | 91.59(+21.59) | 0.0(+0.0) | 0.0(+0.0) | 70.55(+70.55) | 25.66(-14.34) | **96.79(+42.79)** |
| MD-v2  | 0.01(+0.01) | 88.63(+23.43) | 0.0(+0.0) | 0.0(+0.0) | 20.24(+20.24) | 29.14(-16.86) | **96.31(+36.31)** |
| LP-v2  | 0.0(+0.0) | **51.4(+9.6)** | 0.0(+0.0) | 0.0(+0.0) | 0.0(+0.0) | 0.0(+0.0) | 0.0(+0.0) |
| LD-v2  | 0.0(+0.0) | **70.6(+32.6)** | 0.0(+0.0) | 0.0(+0.0) | 0.0(+0.0) | 0.0(+0.0) | 0.0(+0.0) |
| Average | 0.13(+0.07) | **73.83(+12.61)** | 2.65(-15.01) | 4.99(+4.99) | 19.39(+19.39) | 28.97(-0.36) | 62.11(+34.1) |
| Avg. (U+M) | 0.19(+0.08) | 80.26(+8.36) | 3.98(-22.52) | 7.49(+7.49) | 29.08(+29.08) | 43.46(-0.54) | **93.17(+51.17)** |

Table 3: Average reward for each method in Antmaze Environments before and after online finetuning. The best result is marked in bold font and all results $> 90\%$ fo the best performance are underlined. To save space, the name of the environments and datasets are abbreviated as follows: U=Umaze, UD=Umaze-Diverse, MP=Medium-Play, MD=Medium-Diverse, LP=Large-Play and LD=Large-Diverse. U+M=Umaze and Medium maze. Our method performs the best on umaze and medium maze, while IQL performs the best on large maze. Both methods are much better than the rest on average. TD3+BC diverges on antmaze in our experiments.

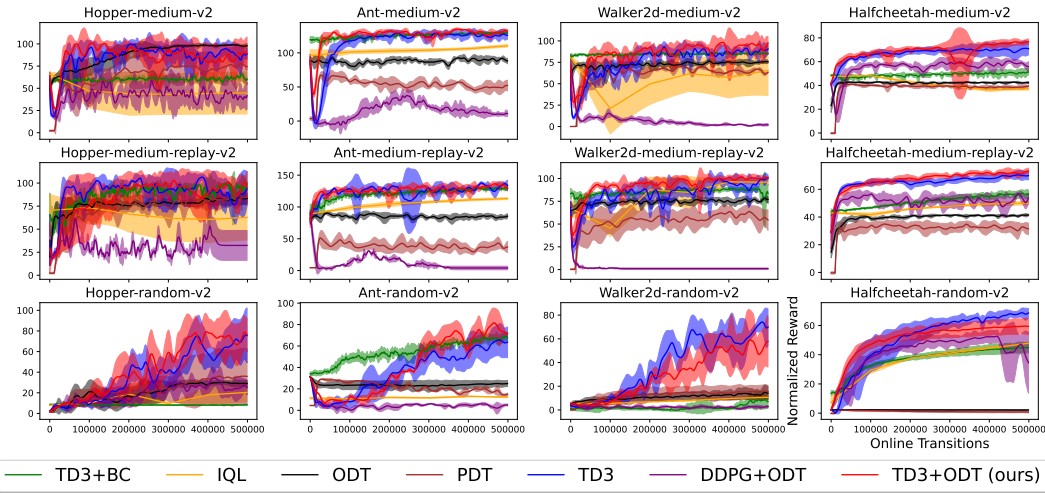

Figure 6: Results on MuJoCo [58] Environments. The TD3 gradient significantly improves the overall performance of the decision transformer; autoregressive algorithms, such as ODT and PDT, fails to improve policy in most cases (especially on random dataset), while TD3+BC and IQL's improvement during finetuning is generally limited.

architecture as ODT which outputs a squashed Gaussian distribution with trainable mean and standard deviation). It is apparent that TD3 [21] and SAC [25] are the RL algorithms that suit the decision transformer best. Fig. 9 further shows the performance comparison between a decision transformer with SAC+ODT mixed gradient and TD3+ODT mixed gradient (both with coefficient $0.1$). The result shows that TD3 is the better choice when paired with supervised learning.

Note, While PPO is generally closely related with transformers (e.g., Reinforcement Learning from Human Feedback (RLHF) [44]), and was used in some prior work for online finetuning of decision transformers *with a small, discrete action space* [38], in our experiments, we find PPO generally does not work with the decision transformer architecture. The main reason for this is the importance sampling issue: PPO has the following objective for an actor $\pi_\theta$ parameterized by $\theta$:

$$\max_\theta \min \left( \mathbb{E}_{(s,a) \sim \pi_{\theta_{\text{old}}}} \frac{\pi_\theta(a|s)}{\pi_{\theta_{\text{old}}}(a|s)} A^{\pi_{\theta_{\text{old}}}}(s,a), \text{clip}\left( \frac{\pi_\theta(a|s)}{\pi_{\theta_{\text{old}}}(a|s)}, 1-\epsilon, 1+\epsilon \right) A^{\pi_{\theta_{\text{old}}}}(s,a) \right). \quad (7)$$

Here, $\pi_{\theta_{\text{old}}}$ is the policy at the beginning of the training for the current epoch. Normally, the denominator of the red part, $\pi_{\theta_{\text{old}}}(a|s)$, would be reasonably large, as the data is sampled from that distribution. However, because of the offline nature caused by different RTGs and context lengths at rollout and training time, the denominator for the red part in Eq. (7) could be very small in training, which will lead to a very small loss if $A^\pi_{\theta_{\text{old}}}(s,a) > 0$. This introduces significant instability during the training process. Fig. 10 illustrates the instability and degrading performance of a PPO-finetuned decision transformer.

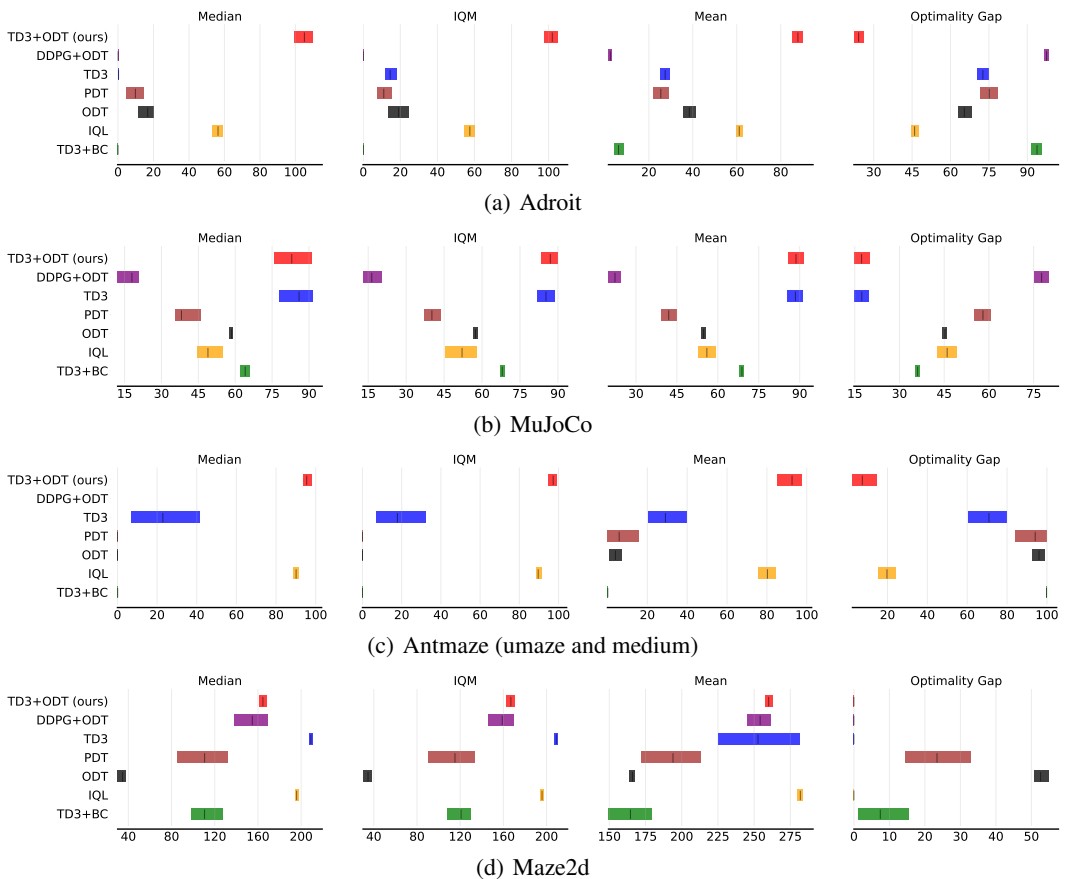

Figure 7: Our main final results re-evaluated using the rliable library with $10000$ bootstrap replications. The x-axes are normalized scores (optimality gap is $\int_0^{100} \Pr(\text{reward} \leq x) dx$). Our method indeed outperforms all baselines on Adroit, MuJoCo and antmaze (umaze and medium).

In contrast, RLHF, does not exhibit such a problem: it does not use different return-to-go and context length in evaluation and training. Thus RLHF does not encounter the problem described above.

Besides the RL gradients mentioned above, as IQL works well on large Antmazes, we also explore the possibility of using IQL as the RL gradient for decision transformer instead of TD3. We found that IQL gradients, when applied to the decision transformer, indeed lead to much better results on antmaze-large. However, IQL fails to improve the policy when the offline dataset subsumes very low reward trajectories, which does not conform with our motivation. This is probably because IQL, as an offline RL algorithm, aims to address out-of-distribution evaluation issue, which is a much more important source of improvement in exploration in the online case. Thus, we choose TD3 as the RL gradient applied to decision transformer finetuning in this work. Fig. 11 shows the result of adding TD3 gradient vs. adding IQL gradient on Antmaze-large-play-v2 and hopper-random-v2.

## D  Why Our Critic Serves as an Average of Policies Generated by Different Context Lengths?

As we mentioned in Sec. 2, When updating a deterministic DT policy, the following loss is minimized:

$$\sum_{t=1}^{T_{\text{train}}} \left\| \mu^{\text{DT}}\left(s_{0:t}, a_{0:t-1}, \text{RTG}_{0:t}, \text{RTG} = \text{RTG}_{\text{real}}, T = t\right) - a_t \right\|_2^2, \tag{8}$$

where $T_{\text{train}}$ is the training context length and $\text{RTG}_{\text{real}}$ is the real return-to-go.

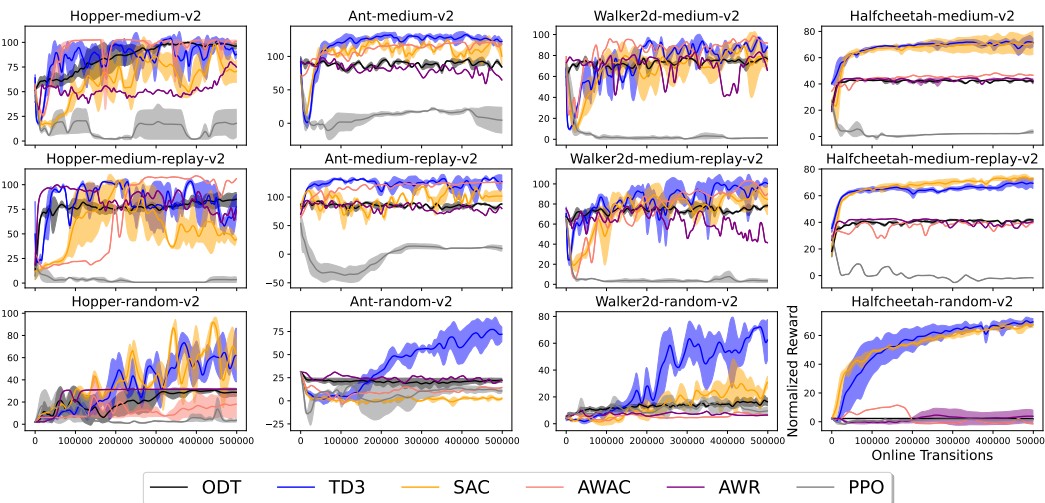

Figure 8: Performance comparison of different, pure RL gradients for online finetuning on standard D4RL benchmarks, with TD3 [21], SAC [25], AWAC [40], AWR [47] and PPO [52]. We also plot ODT's performance as a reference. The result shows that generally, TD3 and SAC work the best, while PPO does not work at all.

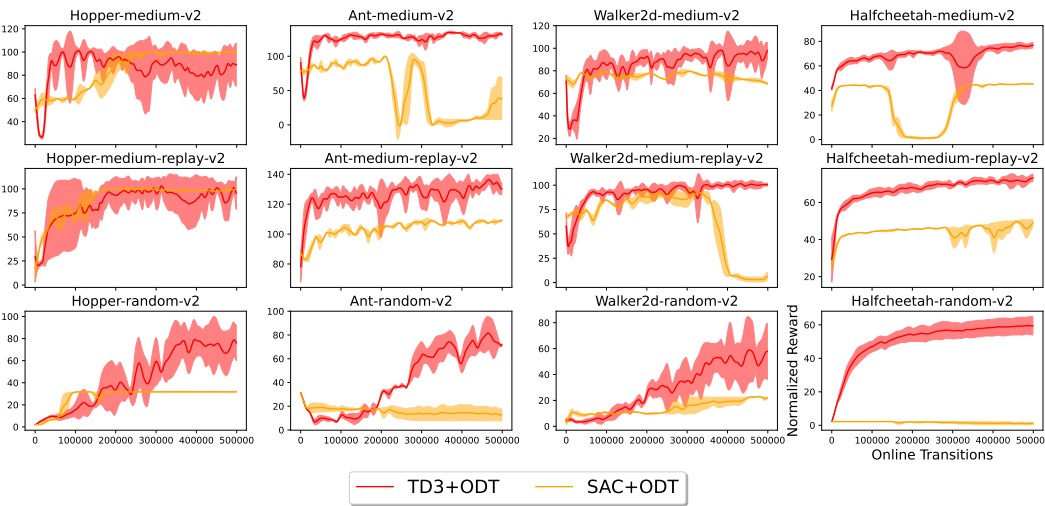

Figure 9: Performance comparison between SAC+ODT and TD3+ODT on standard D4RL benchmarks. TD3+ODT significantly outperforms SAC+ODT.

However, if we consider a particular action $a_t$ in some trajectory $\tau$ of the dataset, during training (both offline pretraining and online finetuning), the policy generated by the decision transformer fitting $a_t$ will be

$$a_t = \mu^{\mathrm{DT}}(s_{t-T:t}, a_{t-T:t-1}, \mathrm{RTG}_{t-T:t}, T \sim U'(1, T_{\mathrm{train}}), \mathrm{RTG} = \mathrm{RTG}_{\mathrm{real}}), \qquad (9)$$

$T$ is actually *sampled* from a distribution $U'(1, T_{\mathrm{train}})$ over integers between $1$ and $T_{\mathrm{train}}$ inclusive; this distribution $U'$ is introduced by the randomized starting step of the sampled trajectory segments, and is *almost* a uniform distribution on integers, except that a small asymmetry is created because the context length will be capped at the beginning of each trajectory. See Fig. 12 for an illustration.

Therefore, online decision transformers (and plain decision transformers) are actually trained to predict with every context length between $1$ and $T_{\mathrm{train}}$. During the training process, the context length is randomly sampled according to $U'$, and a critic is trained to predict an "average value" for the policy generated with context length sampled from $U'$.

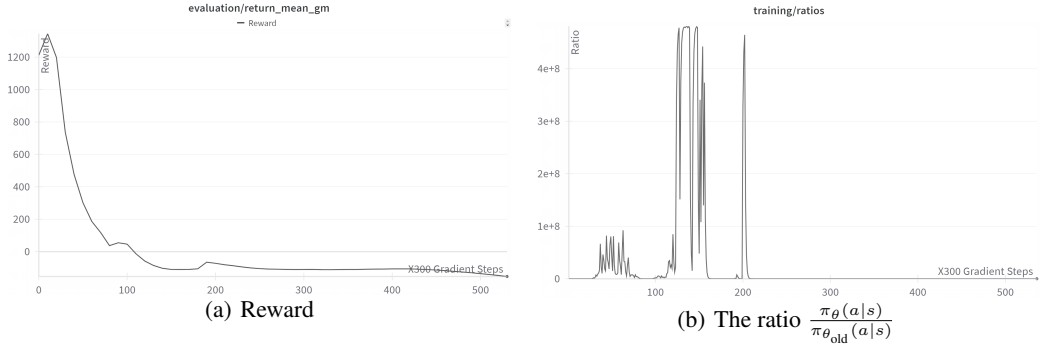

(a) Reward            (b) The ratio $\frac{\pi_\theta(a|s)}{\pi_{\theta_{\text{old}}}(a|s)}$

Figure 10: An illustration of the training instability and the corresponding performance of a PPO-finetuned decision transformer. The $x$ axis is $300\times$ the number of gradient steps.

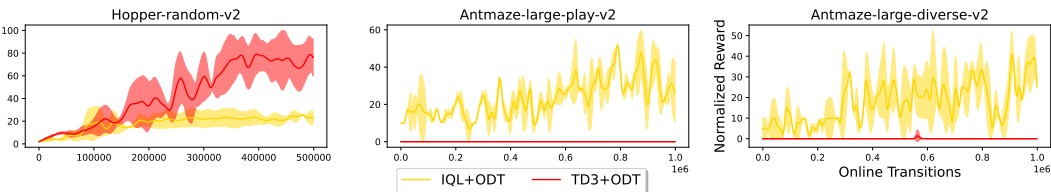

Figure 11: Performance comparison between IQL+ODT and TD3+ODT. While IQL gradient is good at both large antmaze environments, it is much easier to fall into local minima in low-reward offline dataset such as Hopper-random-v2.

## E   Mathematical Proofs

In this section, we will state the theoretical analysis summarized in Sec. 3.3 more rigorously. We will first provide an explanation on how the decision transformer improves its policy during online finetuning, linking it to an existing RL method in Sec. E.1 and Sec. E.2. We will then bound its performance in Sec. E.3.

### E.1   Preliminaries

Advantage-Weighted Actor Critic (AWAC) [40] is an offline-to-online RL algorithm, where the replay buffer is filled with offline data during offline pretraining and then supplemented with online experience during online finetuning. AWAC uses standard $Q$-learning to train the critic $Q : |S| \times |A| \to \mathbb{R}$, and update the actor using weighted behavior cloning, where the weight is exponentiated advantage (i.e., $\exp\left(\frac{A(s,a)}{\lambda}\right)$ where $\lambda > 0$ is some constant).

### E.2   Connection between Decision Transformer and AWAC

We denote $\beta$ as the underlying policy of the dataset, and $P_\beta$ as the distribution over states, actions or returns induced by $\beta$. Note such $P_\beta$ can be either discrete or continuous. By prior work [7], for decision transformer policy $\pi^{\text{DT}}$, we have the following formula holds for any return-to-go RTG $\in \mathbb{R}$ of the future trajectory:

$$\pi^{\text{DT}}(a|s, \text{RTG}) = P_\beta(a|s, \text{RTG}) = \frac{P_\beta(a|s)P_\beta(\text{RTG}|s, a)}{P_\beta(\text{RTG}|s)} = \beta(a|s)\frac{P_\beta(\text{RTG}|s, a)}{P_\beta(\text{RTG}|s)}. \qquad (10)$$

Based on Eq. (10), we have the following lemma:

**Lemma 2.** *For state-action pair $(s, a)$ in an MDP, RTG $\in \mathbb{R}$, assume the distributions of return-to-go (RTG) $P_\beta(RTG|s, a)$ and $P_\beta(RTG|s)$ are Laplace distributions with scale $\sigma$, then for any RTG large*

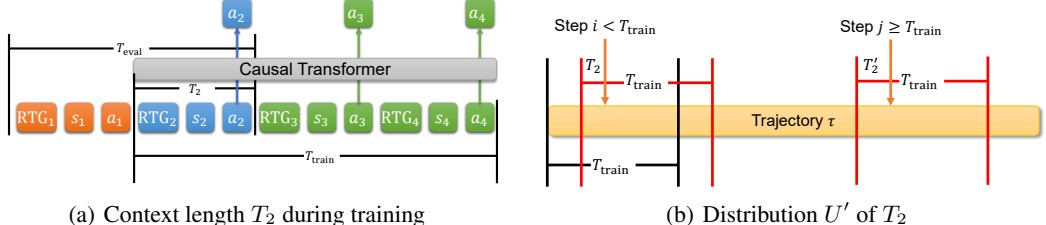

(a) Context length $T_2$ during training

(b) Distribution $U'$ of $T_2$

Figure 12: **a)** illustrates the context length $T_2$ during training; $T_{\text{eval}}$ is the context length of $a_2$ upon sampling and evaluation. It is easy to see that $T_2$ is randomized during training due to the left endpoint of the sampled trajectory segment. **b)** shows the distribution $U'$ of $T_2$; while $T_2'$ for step $j \geq T_{\text{train}}$ is uniformly sampled between 1 and $T_{\text{train}}$ because the start of the segment is uniformly sampled, $T_2$ for step $i < T_{\text{train}}$ will be capped at the start of the trajectory. Thus $U'$ is not exactly uniform.

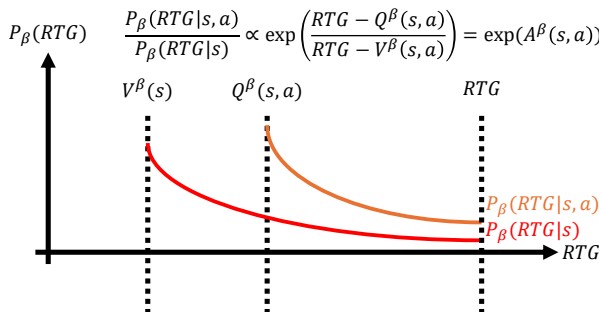

Figure 13: An illustration of how decision transformer policy update at $\text{RTG}_{\text{eval}}$ is related to AWAC. Though the assumption is strong to form the exact same formula, it shows the basic idea of how we link $P^\beta(\text{RTG})$ to its distance to $V^\beta(s)$ and $Q^\beta(s,a)$ in this paper.

*enough (more rigorously, $RTG \geq \max\{Q^\beta(s,a), V^\beta(s)\}$), $\pi(a|s, RTG)$ is updated in the same way as AWAC.*[4]

*Proof.* If return-to-go are Laplace distributions, then by the symmetric property of such distributions, the mean of RTG given $s$ or $(s,a)$ would be the expected future return, which by definition are value functions for $\beta$, i.e., $V^\beta(s)$ for $P_\beta(\text{RTG}|s)$ and $Q^\beta(s,a)$ for $P_\beta(\text{RTG}|s,a)$. See Fig. 13 as an illustration. As $\text{RTG}_{\text{eval}} \geq \max\{Q^\beta(s,a), V^\beta(s)\}$, we have

$$P_\beta(\text{RTG}|s,a) = p_\beta(\text{RTG}|s,a) = \frac{1}{2\sigma}\exp\left(-\frac{\text{RTG} - Q^\beta(s,a)}{\sigma}\right),$$
$$P_\beta(\text{RTG}|s) = p_\beta(\text{RTG}|s,a) = \frac{1}{2\sigma}\exp\left(-\frac{\text{RTG} - V^\beta(s)}{\sigma}\right). \tag{11}$$

And thus we have $\frac{P_\beta(\text{RTG}|s,a)}{P_\beta(\text{RTG}|s)} = \exp\left(\frac{Q^\beta(s,a) - V^\beta(s)}{\sigma}\right) = \exp(\frac{A^\beta(s,a)}{\sigma})$, where $A^\beta$ is the advantage function. $\qquad\square$

While the Laplace distribution assumption in this lemma is strict and impractical for real-life applications, it gives us three crucial insights for a decision transformer:

- For any state $s$ or state-action pair $(s,a)$, we have $\mathbb{E}_\beta[\text{RTG}|s,a] = Q^\beta(s,a)$, $\mathbb{E}_\beta[\text{RTG}|s] = V^\beta(s)$, which is an important property of $P_\beta$ on RTG;

---

[4]Decision transformers have a sequence of past RTGs as input, but the past sequence can be augmented into the state space to fit into such form.

- For decision transformer, the ability to improve the policy by collecting rollouts with high RTG, similar to AWAC, is closely related to advantage. In the above lemma, for example, if the two Laplace distributions have different scales $\sigma_V, \sigma_Q$, we will have the ratio $\frac{P_\beta(\text{RTG}|s,a)}{P_\beta(\text{RTG}|s)}$ being $\exp\left(\frac{Q^\beta(s,a)}{\sigma_Q} - \frac{V^\beta(s)}{\sigma_V}\right)$; if the distributions are Gaussian, we will get similar results but with quadratic terms of $Q^\beta$ and $V^\beta$.

- Different from AWAC, the "policy improvement" of online decision transformers heavily relies on the global property of the return-to-go as $\text{RTG}_{\text{eval}}$ moves further away from $Q^\beta$ and $V^\beta$. If the return-to-go is far away from the support of the data, we will have almost no data to evaluate $P_\beta$, and its estimation can be very uncertain (let alone ratios). In this case, it is very unlikely for the decision transformer to collect rollouts with high $\text{RTG}_{\text{true}}$ and get further improvement. This is partly supported by the corollary of Brandfonbrener et al. [7], where the optimal conditioning return they found on evaluation satisfies $\text{RTG}_{\text{eval}} = V^\beta(s_1)$ at the initial state $s_1$. This is also supported by our single-state MDP experiment discussed in Sec. 3.1 and illustrated in Fig. 2.

Those important insights lead to the intuition that decision transformers, when finetuned online, lack the ability to improve "locally" from low $\text{RTG}_{\text{true}}$ data, and encourages to study the scenario where $P_\beta(\text{RTG}|s)$ and $P_\beta(\text{RTG}|s,a)$ are small.

### E.3 Failure of ODT Policy Update with Low Quality Data

As mentioned in Sec. E.2, we study the performance of decision transformers in online finetuning when $P_\beta(\text{RTG}|s)$ and $P_\beta(\text{RTG}|s,a)$ is small. Specially, in this section, $r(s,a)$ is not a reward function, but a reward **distribution** conditioned on $(s,a)$; $r_i \sim r(s_i, a_i)$ is the reward obtained on $i$-th step. Such notation takes the noise of reward into consideration and forms a more general framework. Also, as the discrete or continuous property of $\beta$ is important in this section, we will use $\text{Pr}_\beta$ to represent probability mass (for discrete distribution or cumulative distribution for continuous distribution) and $p_\beta$ to represent probability density (for probability density function for continuous distribution).

By prior work [7], we have the following performance bound *tight* up to a constant factor for decision transformer for every iteration of updates:

**Theorem E.1.** *For any MDP with transition function $p(\cdot|s,a)$ and reward **random variable** $r(s,a)$, and any condition function $f$, assume the following holds:*

- ***Return coverage:*** $P_\beta(g = f(s_1)|s_1) \geq \alpha_f$ *for any initial state $s_1$;*

- ***Near determinism:*** *for any state-action pair $(s,a)$, $\exists \, s'$ such that $\text{Pr}(s'|s,a) \geq 1 - \epsilon$, and $\exists \, r_0(s,a)$ such that $\text{Pr}(r(s,a) = r_0(s,a)) \geq 1 - \epsilon$;*

- ***Consistency of $f$:*** $f(s) = f(s') + r$ *for all $s$ when transiting to next state $s'$.*

*Then we have*

$$\mathbb{E}_{s_1 \sim p_{ini}}\left[f(s_1)\right] - \mathbb{E}_{\tau = (s_1, a_1, \ldots, s_H, a_H) \sim \pi^{DT}(\cdot|s, f(s))}\left[\sum_{i=1}^{H} \mathbb{E}_{r_i \sim r(s_i, a_i)} r_i\right] \leq \epsilon\left(\frac{1}{\alpha_f} + 2\right) H^2, \quad (12)$$

*where $\alpha_f > 0, \epsilon > 0$ are constants, $p_{ini}$ is the initial state distribution, and $H$ is the horizon of the MDP. $\pi^{DT}$ is the learned policy by Eq. (10).*

*Proof.* See Brandfonbrener et al. [7]. □

In our case, we define $f(s)$ as follows:

**Definition E.2.** $f(s_1) = \text{RTG}_{\text{eval}}$ for all initial states $s_1$, $f(s_{i+1}) = f(s_i) - r_i$ for the $(i+1)$-th step following $i$-th step ($i \in \{1, 2, \ldots, T-1\}$).

Further, we enforce the third assumption in Thm. E.1 by including the cumulative reward so far in the state space (as described in the paper of Brandforbrener et al. [7]). Under such definition, we have a tight bound on the regret between our target $RTG_{eval}$ and the true return-to-go $RTG_{true} = \sum_{i=1}^{H} r_i$ by our learned policy at optimal, based on current replay buffer in online finetuning.

We will now prove that under certain assumptions, $\frac{1}{\alpha_f}$ grows **superlinearly** with respect to RTG; as the bound is tight, the expected cumulative return term $\mathbb{E}_{\tau=(s_1,a_1,\dots,s_H,a_H)\sim\beta}\left[\sum_{i=1}^{H} \mathbb{E}_{r_i\sim r(s_i,a_i)} r_i\right]$ will be decreasing to meet the bounds.

To do this, we start with the following assumptions:

**Assumption E.3.** We assume the following statements to be true:

- (**Bounded reward**) We assume the reward is bounded in $[0, R_{max}]$ for any state-action pairs.

- (**High evaluation RTG**) $RTG_{eval} \geq RTG_{\beta max}$, where $RTG_{\beta max}$ is the largest $RTG_{true}$ in the dataset of $n$ trajectories generated by $\beta$.

- (**Beta prior**) We assign the prior distribution of RTG generated by policy $\beta$ to be a Beta distribution $Beta(1, 1)$ for the binomial likelihood of RTG falling on $[0, RTG_{\beta max}]$ or $[RTG_{\beta max}, TR_{max}]$.

*Remark* E.4. The Beta distribution can be changed to any reasonable distribution; we use Beta distribution only for a convenient derivation. Considering the fact that by common sense, trajectories with high return are very hard to obtain, we can further strengthen the conclusion by changing the prior distribution.

We then prove the following lemma:

**Lemma 3.** *Under the Assumption E.3, given underlying policy $\beta$ of the dataset, for any state $s$ with value function $V^\beta(s)$ and any state-action pair $(s,a)$ with Q-function $Q^\beta(s,a)$, any $c \geq 0$ and $RTG_{eval} \in \mathbb{R}$, with probability $1 - \delta$, we have*

$$
\begin{aligned}
\Pr_\beta(RTG_{eval} - V^\beta(s) \geq c|s) &\leq \frac{(1-\epsilon)RTG_{\beta max} + \epsilon R_{max}^2 T^2 - \left[V^\beta(s)\right]^2}{c^2}, \\
\Pr_\beta(RTG_{eval} - Q^\beta(s,a) \geq c|s,a) &\leq \frac{(1-\epsilon)RTG_{\beta max} + \epsilon R_{max}^2 T^2 - \left[Q^\beta(s,a)\right]^2}{c^2},
\end{aligned}
\tag{13}
$$

*where $\delta = 1 - CDF_{Beta(n+1,1)}(\epsilon)$, and the CDF is the cumulative distribution function.*

*Proof.* With the beta prior assumption in Assumption E.3, we know that with $n$ samples where $RTG \leq RTG_{\beta max}$, we have the posterior distribution to be $Beta(n + 1, 1)$, i.e., with probability $1 - CDF_{Beta(n+1,1)}(\epsilon)$, we have $\Pr_\beta(RTG \geq RTG_{\beta max}) \leq \epsilon$ for $\epsilon > 0$.

Thus, by Chebyshev inequality, we know

$$
\begin{aligned}
\Pr_\beta\left(RTG - V^\beta(s) \geq c|s\right) &\leq \frac{\mathbb{E}_\beta\left[RTG^2\right] - \mathbb{E}_\beta^2\left[RTG\right]}{c^2} \\
&\leq \frac{(1-\epsilon)RTG_{\beta max} + \epsilon R_{max}^2 T^2 - \left[V^\beta(s)\right]^2}{c^2},
\end{aligned}
\tag{14}
$$

and a similar conclusion holds for $\Pr_\beta\left(RTG - Q^\beta(s,a) \geq c|s,a\right)$. Thus, the probability decays superlinearly with respect to RTG. □

Given this lemma, it remains to connect the bound of $\Pr_\beta(RTG \geq c_0)$ to $P_\beta(RTG = c_0)$ on RTG, $c_0 \in \mathbb{R}$. For discrete distribution, the connection is straightforward: $\Pr_\beta(RTG \geq c_0) \geq \Pr_\beta(RTG = c_0) = P_\beta(RTG = c_0)$ for any condition $s$ or $(s,a)$.

Thus, we immediately get the following corollary:

**Corollary 2.** *Assume the reward is bounded in $[0, R_{max}]$ for any state and action, and the number of possible different return-to-go one can get is finite or countably infinite. Then for the $f$ condition function defined in Def. E.2, with probability of at least $1-\delta$, we have $\alpha_f \leq \frac{(1-\epsilon)RTG_{\beta max}+\epsilon R_{max}^2 T^2-[V^\beta(s)]^2}{(RTG-V^\beta(s))^2}$, i.e., $\frac{1}{\alpha_f}$ grows in the order of $\Omega(RTG_{eval}^2)$.*

*Remark* E.5. While the limitation on return-to-go seems strong theoretically, it is very easy to satisfy such assumption in practice because it has no requirement on the discreteness of state and action space. Such corollary can be applied on reward discretization with arbitrary precision (including implicit ones by float precision).

For continuous distribution, to bound $p_\beta(\text{RTG} = c_0)$ with $\Pr_\beta(\text{RTG} \geq c_0)$ on RTG, we would need to assume that "peak" does not exist (see Fig. 14 for illustration), i.e. there does not exist cases where $p_\beta$ is large but $\Pr_\beta$ is small. Thus, we made the following assumption:

**Assumption E.6.** (**Lipschitzness on uncovered RTG distribution**) $p_\beta(\text{RTG}|s)$ is $K_V$-Lipschitz where RTG is larger than $\text{RTG}_{\beta\max}$.

*Remark* E.7. The assumption is reasonable because when RTG is larger than any of the $\text{RTG}_{\text{true}}$ in the dataset, we have no data coverage for the performance of the underlying policy $\beta$ under such RTG, and thus we can choose any inductive bias for $\beta$.

*Remark* E.8. Note the Lipschitzness of $p_\beta$ does not rely on the Lipschitzness of the reward function. For example, consider a single-state, single-step MDP where we have a uniformly random policy $a \sim U(0, 1)$ with reward $r(a) = 2 - \frac{1}{a}$. The reward is clearly not Lipschitz on $a \in (0, 1)$, but the distribution of $r$ is $p_r(r_0) = p_a(r^{-1}(r_0)) \cdot \frac{\partial r^{-1}(r_0)}{\partial r} = \frac{1}{(r-2)^2}$, which is Lipschitz on $(-\infty, 1)$ as $a \in (0, 1)$.

With such assumption, we have the following corollary:

**Corollary 3.** *Under assumption E.6, for the $f$ condition function defined in Def. E.2, we have $\alpha_f \leq \sqrt{2K_V}\Omega\left(RTG_{eval}^{-1.5}\right)$, i.e., $\frac{1}{\alpha_f}$ grows in the order of $\Omega(RTG_{eval}^{1.5})$ with probability of at least $1 - \delta$.*

*Proof.* Under assumption E.6, by Lipschitzness, for any RTG where $p_\beta(\text{RTG}|s) > p_0$, we have $p_\beta(\text{RTG} + c|s) > p_0 - K_V \cdot c$ for any $c \in [0, \frac{p_0}{K_V}]$.

Thus, we know that if $\exists \, \text{RTG}_0 > \text{RTG}_{\beta\max}$ such that $p_\beta(\text{RTG}_0|s) > p_0$, then we have $\Pr_\beta(\text{RTG} \geq \text{RTG}_0|s) > \frac{p_0^2}{2K_V}$ (See Fig. 14 for an illustration). By the contra-positive statement of the above conclusion, we know that

$$\Pr_\beta(\text{RTG} \geq \text{RTG}_0|s) \leq \frac{p_0^2}{2K_V} \Rightarrow \forall \text{RTG} \geq \text{RTG}_0, \, p_\beta(\text{RTG}|s) \leq p_0, \tag{15}$$

and $\text{RTG}_{eval}$ is applicable to the inequality above by the high evaluation RTG assumption in Assumption E.3. We then apply the proof lemma 3, but apply the inequality $P(x - \mathbb{E}[x] \geq c) \leq \frac{\mathbb{E}(x-\mathbb{E}[x])^3}{c^3}$ instead of Chebyshev inequality which leads to the conclusion. $\qquad\square$

# F  Experimental Details

## F.1  Environment and Dataset Details

### F.1.1  Single-State MDP

The single-state MDP studied in Sec. 3.1 motivates why RL gradients are useful for online finetuning. It has a single state, a single action $a \in [-1, 1]$, and a reward function $r(a) = (a + 1)^2$ if $a \leq 0$ and $r(a) = 1 - 2a$ otherwise.

**Datasets.** The dataset has a size of 128, with 100 actions uniformly sampled in $(-1, 0.95)$, and the remaining 28 actions uniformly sampled in $(0.5, 1)$. The dataset is designed to conceal the reward peak in the middle. DDPG and ODT+DDPG successfully recognized the reward peak but ODT failed.

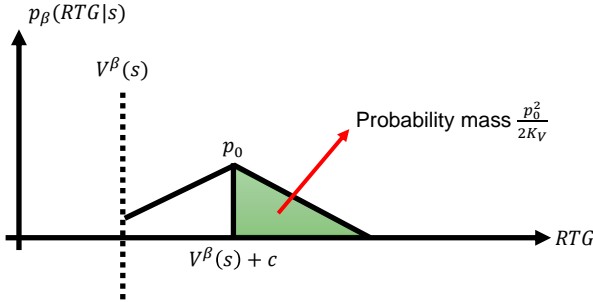

Figure 14: An illustration of how Lipschitzness on the distribution of $p_\beta(\text{RTG}|s)$ could link the bound between $\Pr(\text{RTG} \geq V^\beta(s) + c|s)$ and $p_\beta(\text{RTG}|s)$. Note we do not take the left-hand side probability mass of $p_0$ into account because the triangle of probability mass could be truncated by $V^\beta(s)$.

### F.1.2 Adroit Environments

**Environments.** Adroit is a set of more difficult benchmark than Mujoco in D4RL, and is becoming increasingly popular in recent offline and offline-to-online RL works [28, 23]. We test four environments in adroit in our experiments: pen, hammer, door and relocate. Fig. 15 shows an illustration of the four environments.

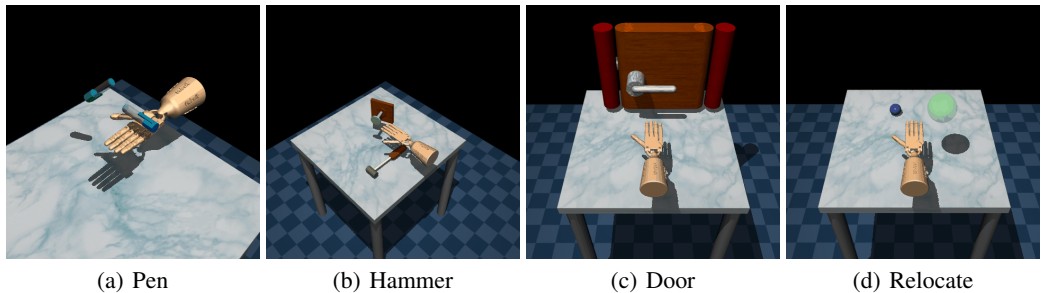

(a) Pen      (b) Hammer      (c) Door      (d) Relocate

Figure 15: Illustration of Adroit environments used in Sec. 4 based on OpenAI Gym [8] and D4RL [19].

1. **Pen.** Pen is a locomotion environment where the agent needs to control a robotic hand to manipulate a pen, such that its orientation matches the target. It has a 24-dimensional action space, each of which controls a joint on the wrist or fingers. The state space is 45-dimensional, which contains the pose of the palm, the angular position of the joints, and the pose of the target and current pen.

2. **Hammer.** Hammer is an environment where the agent needs to control a robotic hand to pick up a hammer and use it to drive a nail into a board. The action space is 26-dimensional, each of which corresponds to a joint on the hand. The state space is 46-dimensional, which describes the angular position of the fingers, the pose of the palm, and the status of hammer and nail.

3. **Door.** In the door environment, the agent needs to use a robotic hand to open a door by undoing the latch and swinging it. The environment has a 28-dimensional action space, which are the absolute angular positions of the hand joints. It also has 39-dimensional observation space which describes each joint, the pose of the palm, and the door with its latch.

4. **Relocate.** In the relocate environment, the agent needs to control a robotic hand to move a ball from its initial location towards a goal, both of which are randomized in the environment. The environment has a 30-dimensional action space which describes the angular position of the joints on the hand, and a 39-dimensional space which describes the hand as well as the ball and target.

| Dataset | Size | Normalized Reward |
|---|---|---|
| Pen-expert-v1 | 499106 | $107.40 \pm 55.65$ |
| Pen-cloned-v1 | 499886 | $108.63 \pm 122.43$ |
| Pen-human-v1 | 4800 | $202.69 \pm 154.48$ |
| Hammer-expert-v1 | 999800 | $96.95 \pm 50.65$ |
| Hammer-cloned-v1 | 999872 | $8.11 \pm 23.35$ |
| Hammer-human-v1 | 10948 | $23.80 \pm 33.36$ |
| Door-expert-v1 | 999800 | $101.19 \pm 16.31$ |
| Door-cloned-v1 | 999939 | $12.29 \pm 18.35$ |
| Door-human-v1 | 6504 | $28.35 \pm 13.88$ |
| Relocate-expert-v1 | 999800 | $102.25 \pm 19.83$ |
| Relocate-cloned-v1 | 999724 | $28.99 \pm 42.88$ |
| Relocate-human-v1 | 9614 | $87.22 \pm 21.28$ |

Table 4: The size and the average and standard deviation of the normalized reward of the Adroit datasets from D4RL [19] used in our experiments.

**Datasets.** For each of the four environments, we test our method across three different qualities of datasets: expert, cloned and human, all of which provided by the DAPG [49] repository. The expert dataset is generated by a fine-tuned RL policy; the cloned dataset is collected from an imitation policy on the demonstrations from the other two datasets; and the human dataset is collected from human demonstrations. Tab. 4 shows the size and average reward of each dataset.

### F.1.3 Antmaze Environments

**Environments.** Antmaze is a more difficult version of Maze2D, where the agent controls a robotic ant instead of a point mass through the maze. It has a 27 dimensional-state space and a 8-dimensional action space. We test our method on six variants of antmaze: Umaze, Umaze-Diverse, Medium-Play, Medium-Diverse, Large-Play and Large-Diverse, where "Umaze", "Medium" and "Large" describes the size of the maze (see Fig. 16 for an illustration), and the "Diverse" and "Play" describes the type of the dataset. More specifically, "Diverse" means that in the offline dataset, the starting point and the goal of the agent are randomly generated, while "Play" means that the goal is generated by a handcraft design. "Umaze" without suffix is the simplest environment where both the starting point and the goal are fixed.

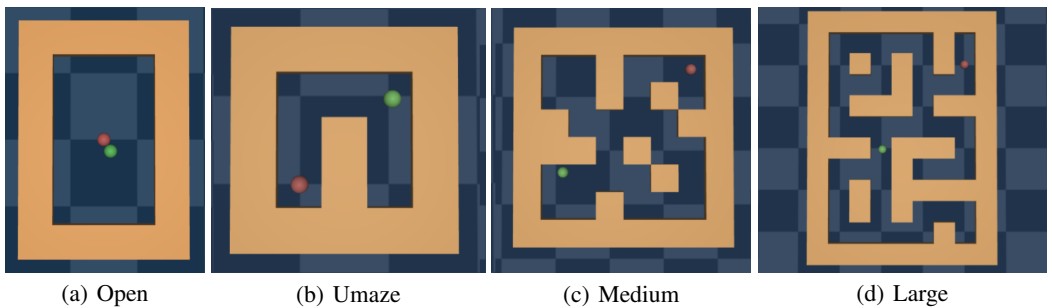

| (a) Open | (b) Umaze | (c) Medium | (d) Large |

Figure 16: Illustration of mazes in antmaze and maze2d environment, where the red point is the goal and the green point is the current location of the agent.

**Datasets.** Similar to Adroit and MuJoCo, we test our method on datasets provided by D4RL. Tab. 5 shows the size and normalized reward of each dataset. Note, following IQL [28] and CQL [29], we conduct reward shaping: subtracting from all rewards in the dataset and environment the value 1 during training of both our method and baselines to provide denser reward signal for all antmaze environments. However, we still count original sparse reward when comparing the performance.

### F.1.4 MuJoCo

**Environments.** We test our method on four widely used environments: Hopper, Halfcheetah, Walker2d and Ant. Fig. 17 shows an illustration of the four environments.

| Dataset | Size | Normalized Reward |
|---|---|---|
| Antmaze-Umaze-v2 | 998573 | $86.14 \pm 34.55$ |
| Antmaze-Umaze-Diverse-v2 | 999000 | $3.48 \pm 18.32$ |
| Antmaze-Medium-Play-v2 | 999000 | $90.85 \pm 28.83$ |
| Antmaze-Medium-Diverse-v2 | 999000 | $66.29 \pm 47.27$ |
| Antmaze-Large-Play-v2 | 999000 | $92.73 \pm 25.97$ |
| Antmaze-Large-Diverse-v2 | 999000 | $86.17 \pm 34.52$ |

Table 5: The size and the average and standard deviation of the normalized reward of the Antmaze datasets from D4RL [19] used in our experiments.

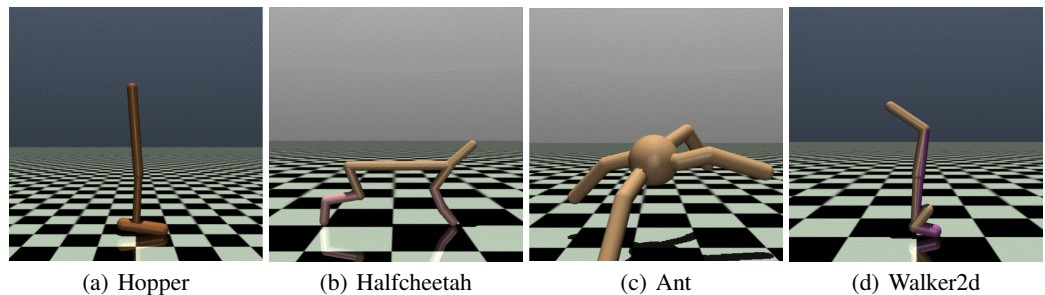

(a) Hopper  (b) Halfcheetah  (c) Ant  (d) Walker2d

Figure 17: Illustration of MuJoCo environments used in Sec. 4 based on OpenAI Gym [8] and D4RL [19].

1. **Hopper.** Hopper is a locomotion task on a 2D vertical plane, where the agent manipulates a single-legged robot to hop forward. Its state is 11-dimensional, which describes the angle and velocity for the robot's joints. Its action is 3-dimensional, which corresponds to the torques applied on the three joints for the current time step respectively.

2. **Halfcheetah.** Halfcheetah is also a 2D environment which requires the agent to control a cheetah-like robot to run forward. The states are 17-dimensional, containing the coordinate and velocity of the joints The actions are 6-dimensional, which control the torques on the joints of the robot.

3. **Ant.** In Ant, the agent controls a four-legged 8-DoF robotic ant to walk in a 3D environment and tries to move forward. It has a 111-dimensional state space describing the coordinates and velocities of the joints.

4. **Walker2d.** Walker2d is a 2D environment in which the agent needs to manipulate a 8-DoF two-legged robot to walk forward under the agent's control. Its state space is 27-dimensional.

**Datasets.** We test our method across three different qualities of datasets: medium, medium-replay and random. The medium dataset contains trajectories collected by an agent trained with RL, but early-stopped at medium-level performance. The medium-replay dataset is the collection of trajectories sampled in the training process of the agent mentioned above. The random dataset contains trajectories collected by an agent with random policy. Tab. 6 shows the size and normalized reward of each dataset.

### F.1.5 Maze2D Environments

**Environments.** Maze2D is another set of D4RL environment, where the agent needs to control a point mass to navigate through a 2D maze and arrive at a fixed goal. It has a 4-dimensional state space describing its coordinate and velocity, and a 2-dimensional action describing its acceleration. The reward is determined by its current distance to the goal. We test our method on four variants of maze: Open, Umaze, Medium and Large with increasing difficulty. The map of each maze is illustrated in Fig. 16. Maze2D environment is tested in Sec. G.2.

**Datasets.** We again test our method on datasets provided by D4RL. Tab. 7 shows the size and normalizeed reward of each dataset.

| Dataset | Size | Normalized Reward |
|---|---|---|
| Hopper-medium-v2 | 999906 | $44.32 \pm 12.27$ |
| Hopper-medium-replay-v2 | 402000 | $14.98 \pm 16.32$ |
| Hopper-random-v2 | 999996 | $1.19 \pm 1.16$ |
| HalfCheetah-medium-v2 | 1000000 | $40.68 \pm 5.12$ |
| HalfCheetah-medium-replay-v2 | 202000 | $27.17 \pm 15.79$ |
| HalfCheetah-random-v2 | 1000000 | $0.07 \pm 2.90$ |
| Walker2d-medium-v2 | 999995 | $62.09 \pm 23.83$ |
| Walker2d-medium-replay-v2 | 302000 | $14.84 \pm 19.48$ |
| Walker2d-medium-random-v2 | 999997 | $0.01 \pm 0.09$ |
| Ant-medium-v2 | 999946 | $80.30 \pm 35.82$ |
| Ant-medium-replay-v2 | 302000 | $30.95 \pm 31.66$ |
| Ant-medium-random-v2 | 999930 | $6.36 \pm 10.07$ |

Table 6: The size and the average and standard deviation of the normalized reward of the MuJoCo datasets from D4RL [19] used in our experiments.

| Dataset | Size | Normalized Reward |
|---|---|---|
| Maze2D-Open-v0 | 999999 | $30.08 \pm 50.17$ |
| Maze2D-Umaze-v1 | 999869 | $-12.55 \pm 9.82$ |
| Maze2D-Medium-v1 | 1999733 | $-3.46 \pm 3.95$ |
| Maze2D-Large-v1 | 3999692 | $-1.71 \pm 2.87$ |

Table 7: The size and the average and standard deviation of the normalized reward of the Maze2D datasets from D4RL [19] used in our experiments.

## F.2 Hyperparameters

### F.2.1 Single-State MDP

For all networks, we use a simple MDP with two hidden layers of width 128, and ReLU [1] as activation function. We add a Tanh activation function to limit the output for ODT and the actor of DDPG to $[-1, 1]$. For both methods, we use Adam [27] as the optimizer, and the learning rate is set to $10^{-3}$. We pretrain 5 epochs on offline data (20 gradient steps) and 16 epochs for online finetuning, with a batch size of 32 for gradient update and collect 64 new rollout states for each epoch (thus we train $2n + 4$ steps for the $n$-th online finetuning epoch). $\text{RTG}_{\text{eval}}$ is set at 1, which serves as the (constant) input for ODT rollout and DDPG actor. Both DDPG and ODT uses deterministic actor with an exploration noise uniform in $[-0.01, 0.01]$ during online rollouts.

### F.2.2 Other Experiments

Tab. 8 summarizes hyperparameters that are common across all environments, and Tab. 9 summarizes hyperparameters that are different across environments. For environments that exist in ODT [74], we follow the hyperparameters from ODT medium environments. We did not use positional embedding as suggested by ODT [74]. Specially, for antmaze, we remove most (all but 10) 1-step trajectories, because the size of the replay buffer for decision transformers is controlled by the number of trajectories, and antmaze dataset contains a large number of 1-step trajectories due to its data generation mechanism (immediately terminate an episode when the agent is close to the goal, but do not reset the agent location). Also, we add Layernorm [4] after each hidden layer of the critic for Adroit, Maze and Antmaze environments, according to Yue et al. [71]'s advice. We found that such practice stabilizes the training process (see Sec. G.5 for ablation).

For ODT and TD3 baseline, we use the same code as our TD3+ODT, while setting coefficients for RL and supervised gradients accordingly. For PDT baseline, we use the default hyperparameter in PDT paper, and pretrain PDT for 40K steps for all experiments. For TD3+BC and IQL, we use the default hyperparameter in their codebase, and pretrain them for 1M steps for all experiments (remaining the same as that in the codebase).

| Hyperparam | Value |
|---|---|
| # dim of embedding dimensions | 512 |
| # of attention heads per layer | 4 |
| # of transformer layers | 4 |
| Dropout | 0.1 |
| Actor Optimizer | LAMB [68] |
| # steps collected per epoch | $\geq 1000$ (random dataset), 1 trajectory (others) |
| Actor activation function | ReLU |
| Scheduler | $10^4$ steps, linear warmup |
| Critic layer | 2 |
| Critic width | 256 |
| Critic activation function | ReLU |
| Batch size | 256 |
| # actor update per epoch | 300 |
| Online exploration noise | 0.1 |
| TD3 policy noise | 0.2 |
| TD3 noise clip | 0.5 |
| TD3 target update ratio | 0.005 |

Table 8: The common hyperparameters across all environments used in our experiments.

| | $T_{\text{train}}$ | $T_{\text{eval}}$ | RTG$_{\text{eval}}$ | RTG$_{\text{online}}$ | Pretrain $\alpha$ | Online $\alpha$ | $\gamma$ | $lr_c$ | $lr_a$ | Weight decay | # pretrain steps | Buffer size |
|---|---|---|---|---|---|---|---|---|---|---|---|---|
| Hopper | 20 | 5 | 3600 | 7200 | 0 | 0.1 | 0.99 | $10^{-3}$ | $10^{-4}$ | 0.0005 | 5K | 1K |
| HalfCheetah | 20 | 5 | 6000 | 12000 | 0 | 0.1 | 0.99 | $10^{-3}$ | $10^{-4}$ | $10^{-4}$ | 5K | 1K |
| Walker2d | 20 | 5 | 5000 | 10000 | 0 | 0.1 | 0.99 | $10^{-3}$ | $10^{-3}$ | $10^{-3}$ | 10K | 1K |
| Ant | 20 | 1 | 6000 | 12000 | 0 | 0.1 | 0.99 | $10^{-3}$ | $10^{-3}$ | $10^{-4}$ | 5K | 1K |
| Pen | 5 | 1 | 12000 | 12000 | 0 | 0.1 | 0.99 | 0.0002 | $10^{-4}$ | $10^{-4}$ | 40K | 5K |
| Hammer | 5 | 5 | 16000 | 16000 | 0 | 0.1 | 0.99 | 0.0002 | $10^{-4}$ | $10^{-4}$ | 40K | 5K |
| Door | 5 | 1 | 4000 | 4000 | 0 | 0.1 | 0.99 | 0.0002 | $10^{-4}$ | $10^{-4}$ | 40K | 5K |
| Relocate | 5 | 1 | 5000 | 5000 | 0 | 0.1 | 0.99 | 0.0002 | $10^{-4}$ | $10^{-4}$ | 40K | 5K |
| Maze2D-Open | 1 | 1 | 120 | 120 | 0 | 0.1 | 0.99 | 0.0002 | $10^{-4}$ | $10^{-4}$ | 40K | 5K |
| -Umaze | 1 | 1 | 60 | 60 | 0 | 0.1 | 0.99 | 0.0002 | $10^{-4}$ | $10^{-4}$ | 40K | 2.5K |
| -Medium | 1 | 1 | 60 | 60 | 0 | 0.1 | 0.99 | 0.0002 | $10^{-4}$ | $10^{-4}$ | 40K | 5K |
| -Large | 5 | 1 | 60 | 60 | 0 | 0.1 | 0.99 | 0.0002 | $10^{-4}$ | $10^{-4}$ | 40K | 5K |
| Antmaze-Umaze | 5 | 1 | -100 | -100 | 0.1 | 0.1 | 0.998 | 0.0002 | $10^{-4}$ | $10^{-4}$ | 40K | 2K |
| -Medium | 1 | 1 | -200 | -200 | 0.1 | 0.1 | 0.998 | 0.0002 | $10^{-4}$ | $10^{-4}$ | 200K | 2K |
| -Large | 5 | 1 | -500 | -500 | 0.1 | 0.1 | 0.998 | 0.0002 | $10^{-4}$ | $10^{-4}$ | 200K | 2K |

Table 9: Environment-specific hyperparameters, where $T_{\text{train}}$ and $T_{\text{eval}}$ stands for training and evaluation context length, RTG$_{\text{eval}}$ and RTG$_{\text{online}}$ represents RTG during evaluation and online rollout respectively, $\alpha$ is the coefficient for RL gradient, $\gamma$ is the discount factor, $lr_c$ is the critic learning rate, and $lr_a$ is the actor learning rate. Buffer size is counted in the number of trajectories. Note RTGs of antmaze have been modified according to our reward shaping.

# G  More Results

## G.1  Delayed Reward

Though we have tested MuJoCo environments in Sec. 4, it is worth noting that many offline RL algorithms have addressed the MuJoCo benchmark quite well [28, 23]. Thus, we also tested settings where RL struggles to obtain good performance to further analyze the performance of using RL gradients for decision transformers.

**Environment and Experimental Setup.** In this experiment, we use the same experiment and dataset as in Sec. 4 except for one major difference: the rewards are not given immediately after each step. Instead, the cumulative reward during a short period of $M$ steps is given *only at the end of the period*, while the rewards observed by the agents within a period are all zero. We adopt such a setting from prior influential work [43], which creates a sparse-reward setting where RL algorithms struggle. We test $M = 5$ in this experiment.

**Results.** We use the same baselines as that in Sec. 4; Tab. 10 summarizes the performance of each method. Generally, DT with TD3 gradient still works very well, much better than ODT. While

|          | TD3+BC | IQL | ODT | PDT | TD3 | TD3+ODT (ours) |
|----------|--------|-----|-----|-----|-----|----------------|
| Ho-M-v2  | 50.2(-1.43) | 43.09(-21.28) | **92.9(+43.03)** | 83.56(+81.73) | 82.11(+17.59) | 82.91(+18.61) |
| Ho-MR-v2 | **99.85(+42.33)** | 79.63(-4.72) | 85.07(+67.6) | 82.2(+80.21) | 90.81(+48.79) | 98.55(+67.1) |
| Ho-R-v2  | 8.8(+0.3) | 18.49(+10.62) | 30.58(+28.37) | 25.1(+23.86) | **75.55(+73.57)** | 52.38(+50.4) |
| Ha-M-v2  | 50.45(+2.56) | 40.31(-6.88) | 42.07(+20.49) | 38.71(+38.66) | 65.92(+24.91) | **66.33(+26.1)** |
| Ha-MR-v2 | 53.59(+8.82) | 47.01(+3.46) | 39.45(+22.36) | 32.55(+32.76) | **57.42(+28.48)** | 49.95(+20.23) |
| Ha-R-v2  | 42.11(+28.73) | 34.87(+28.34) | 2.16(-0.07) | -0.85(+0.96) | **52.24(+49.99)** | 48.79(+46.54) |
| Wa-M-v2  | 84.73(+2.19) | 62.01(-16.59) | 76.21(+3.65) | 59.79(+59.52) | 86.42(+18.88) | **87.83(+19.57)** |
| Wa-MR-v2 | 61.1(-11.8) | 86.67(+13.93) | 76.96(+5.25) | 53.73(+37.17) | **96.38(+34.42)** | 91.93(+27.92) |
| Wa-R-v2  | 6.78(+4.37) | 7.13(+1.22) | 7.93(+3.73) | 18.35(+18.2) | **57.43(+53.32)** | 56.72(+51.72) |
| An-M-v2  | 114.21(+18.09) | 103.34(+6.4) | 80.29(-0.83) | 49.89(+45.92) | **118.21(+30.11)** | 110.67(+23.75) |
| An-MR-v2 | **122.11(+30.56)** | 103.5(+24.91) | 84.61(-2.13) | 42.58(+39.17) | 112.2(+23.98) | 116.6(+28.59) |
| An-R-v2  | **77.41(+21.34)** | 10.77(+0.3) | 22.55(-8.82) | 17.37(+13.69) | 49.3(+17.87) | 53.06(+21.54) |
| Average  | 64.28(+12.17) | 53.07(+3.31) | 53.4(+15.22) | 41.92(+39.32) | **78.68(+35.15)** | 76.31(+33.51) |

Table 10: Average reward for each method in MuJoCo Environments before and after online finetuning with delayed rewards. To save space, the name of the environments and datasets are compressed, where Ho=Hopper, Ha=HalfCheetah, Wa=Walker2d, An=Ant for the environment, and M=Medium, MR=Medium-Replay, R=Random for the dataset. The format is "final(+increase)". The best result for each setting is highlighted in bold font, and any result $> 90\%$ of the best performance is underlined.

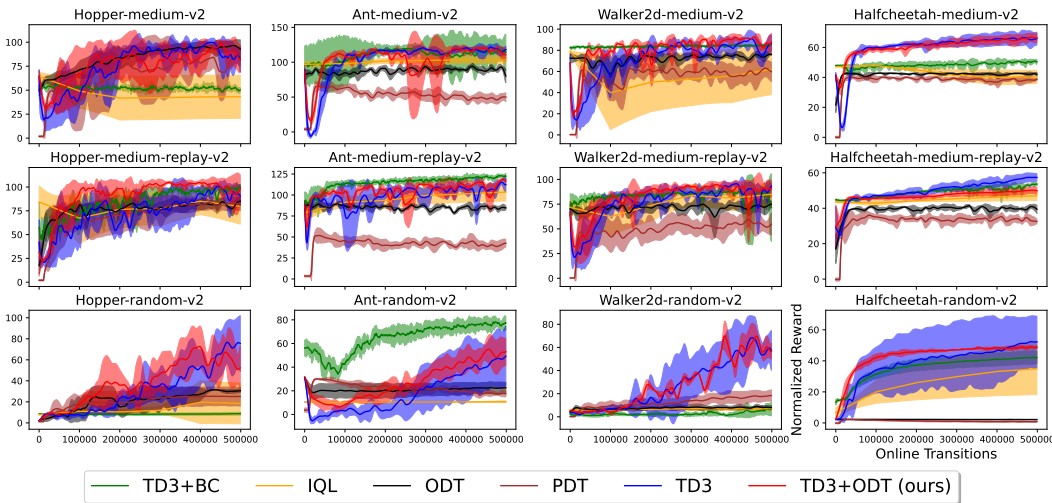

Figure 18: Reward curves for MuJoCo environments with delayed reward.

TD3+BC works well in several scenarios, it struggles on random environments. See Fig. 18 for reward curves.

## G.2 Maze2D Environment

We test on navigation tasks in D4RL [19] where the agents need to control a pointmass through four different mazes: Open, Umaze, Medium and Large with different dataset respectively. Fig. 19 lists the performance of each method on Maze2D before and after online finetuning, and Tab. 11 summarizes the performance before and after online finetuning. The result shows that our method again significantly outperforms autoregressive-based algorithms such as ODT and PDT, which validates our motivation in Sec. 3.1. DDPG+ODT works similarly well as TD3+ODT in this environment with simple state and action space.

|           | TD3+BC | IQL | ODT | PDT | TD3 | DDPG+ODT | TD3+ODT (ours) |
|-----------|--------|-----|-----|-----|-----|----------|----------------|
| Open-v0   | 350.58(-0.22) | **576.06(+24.58)** | 574.4(+306.56) | 515.97(+485.99) | 430.61(-73.36) | 574.03(+96.02) | 574.32(+29.57) |
| Umaze-v2  | 90.91(+63.15) | 159.97(+114.77) | 43.19(+6.37) | 140.84(+153.47) | **162.04(+130.44)** | 139.45(+113.4) | 140.1(+107.76) |
| Medium-v2 | 96.91(+63.2) | 177.61(+97.31) | 26.11(+12.77) | 80.36(+81.49) | **184.55(+138.57)** | 133.03(+95.32) | 136.63(+91.69) |
| Large-v2  | 132.66(+38.21) | 214.02(+143.21) | 20.37(+17.25) | 38.74(+39.14) | **232.9(+218.72)** | 170.09(+151.44) | 189.71(+170.17) |
| Average   | 167.77(+41.09) | **281.92(+94.97)** | 166.02(+85.74) | 193.98(+190.02) | 252.53(+103.59) | 254.15(+114.05) | 260.19(+99.8) |

Table 11: Average reward for each method in Maze2D Environments before and after online finetuning. Our method works slightly worse than IQL but better than all other baselines.

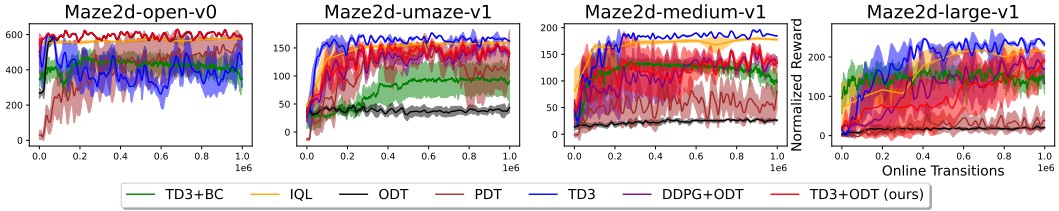

Figure 19: Reward curve for each method in Maze2D Environments. Again, autoregressive algorithms such as ODT and PDT does not perform well in this case.

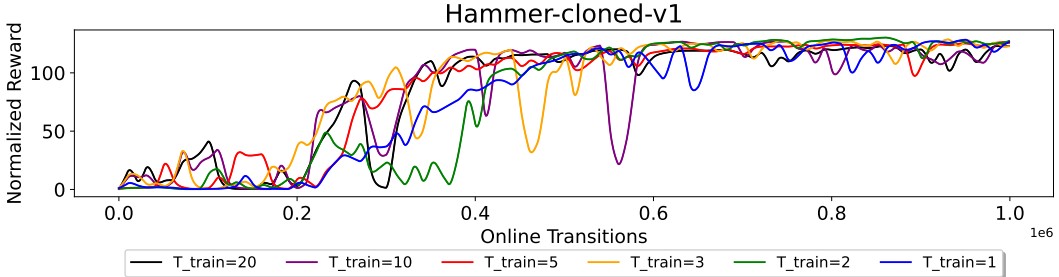

Figure 20: The reward curves on hammer-cloned-v1 with different $T_{\text{train}}$ and $T_{\text{eval}} = 1$. While longer $T_{\text{train}}$ leads to faster convergence in this environment, runs with too long $T_{\text{train}}$ are also unstable.

### G.3 Ablations on training context length $T_{\text{train}}$

Fig. 20 shows the result of using different context lengths on hammer-cloned-v1 environment (in this experiment, we use $T_{\text{eval}} = 1$ to demonstrate the effect of more different $T_{\text{train}}$). It is shown from the experiment that the selection of $T_{\text{train}}$ needs to be balanced between more information taken into account and training stability; while longer $T_{\text{train}}$ brings faster convergence when growing from 1 to 5, the reward curves with $T_{\text{train}} \in \{10, 20\}$ oscillates more than that with $T_{\text{train}} = 5$.

### G.4 Longer Training Process

In some environments, such as hopper-random-v2, walker2d-random-v2 and ant-random-v2, our proposed method still seems to be improving after 500K online samples. In Fig. 21, we show the finetuning result of our proposed method with more online transitions, which effectively shows that our method has greater potential in online finetuning when finetuned for more gradient steps.

### G.5 The Effect of Layernorm

As we have mentioned in Sec. F.2, as suggested by Yue et al. [71], we apply Layernorm [4] to critic networks for environment other than MuJoCo for better stability in training. In our experiment, we found that it greatly stabilizes the critic on complicated environments such as Adroit, but makes online policy improvement less efficient on easier MuJoCo environments. Fig. 22 shows the performance and critic MSE loss comparison on some environments with and without layernorm; it is clearly shown that layernorm helps stabilizes online finetuning in some cases such as pen-cloned-v1, but hinders performance increase on other environments such as halfcheetah-medium-v2.

### G.6 Recurrent Critic

As mentioned in Sec. 3.2, we use reflexive critics (i.e., critics that only take the current state nand action) to add RL gradients to decision transformer, and this creates an average effect among policies generated by different context lengths (see Sec. D in the appendix for detail). In this section, we explore recurrent critic by substituting the MLP critic using a LSTM, such that for a trajectory segment, the evaluation for the $i$-th action $a_i$ is based on all state-action pairs $(s_1, a_1), \ldots, (s_{i-1}, a_{i-1})$ and current state $s_i$. As shown in Fig. 23, we found that recurrent critics are much less stable than

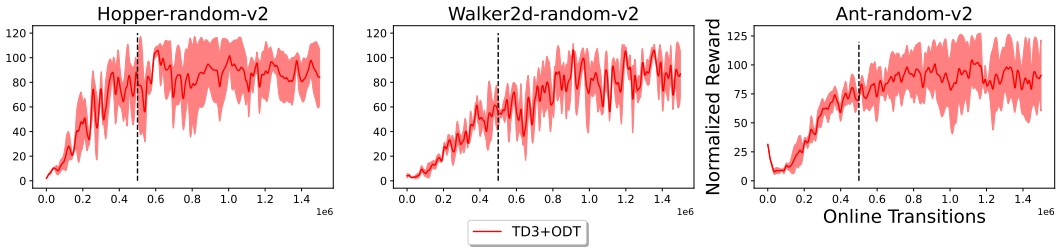

Figure 21: The reward curves of our method when finetuned for more steps (we only report curves until the black line in Fig. 6). It is clearly shown that our method has greater potential for improvement when finetuned for more steps.

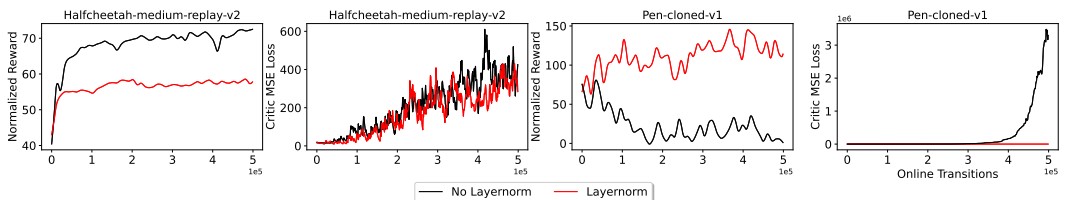

Figure 22: The reward curve and critic MSE loss comparison between runs with and without layer-norm. Layernorm effectively stabilizes online finetuning in pen-cloned-v1, but hinders performance increase in halfcheetah-medium-v2.

reflexive critics, and the instability increases as the training context length $T_{\text{train}}$ grows; on the contrary, reflexive critic can well-handle the case where $T_{\text{train}}$ is long.

### G.7 Regularizer for Pure TD3 Gradients

In the Adroit environment results discussed in Sec. 4, we found that the baseline of ODT finetuned using pure TD3 gradients struggles due to catastrophic forgetting. Inspired by Wołczyk et al. [61], we test whether adding a KL regularizer can fix the forgetting problem. Though our policy is deterministic, we can approximately interpret the policy as Gaussian with a very small variance. Thus, a KL regularizer can be simply added using $c_0 \cdot \|a - a_{\text{old}}\|^2$, where $a$ is the current action and $a_{\text{old}}$ is the action predicted by the pretrained policy. We set $c_0 = 0.05$ and test this method on the Adroit cloned and expert dataset. We illustrate the result in Fig. 24. We find that the KL regularizer effectively addresses the issue on expert environments for both TD3 and TD3+ODT. But it can sometimes hinder the policy improvement of TD3+ODT with low return during online finetuning.

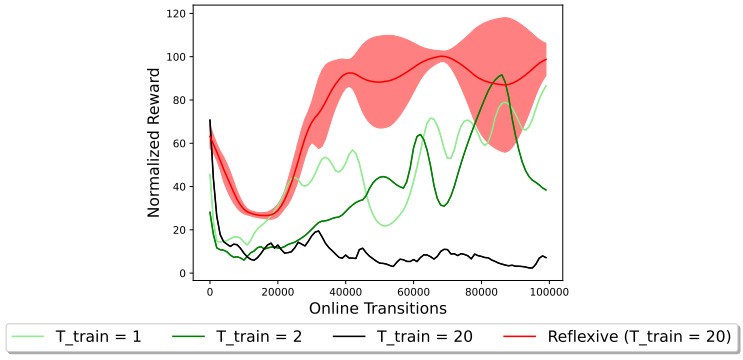

Figure 23: The performance of reflexive critic vs. recurrent critic on hopper-medium-v2. It is clearly shown that recurrent critic is much harder to train, and its performance decreases as the training context length $T_{\text{train}}$ grows.

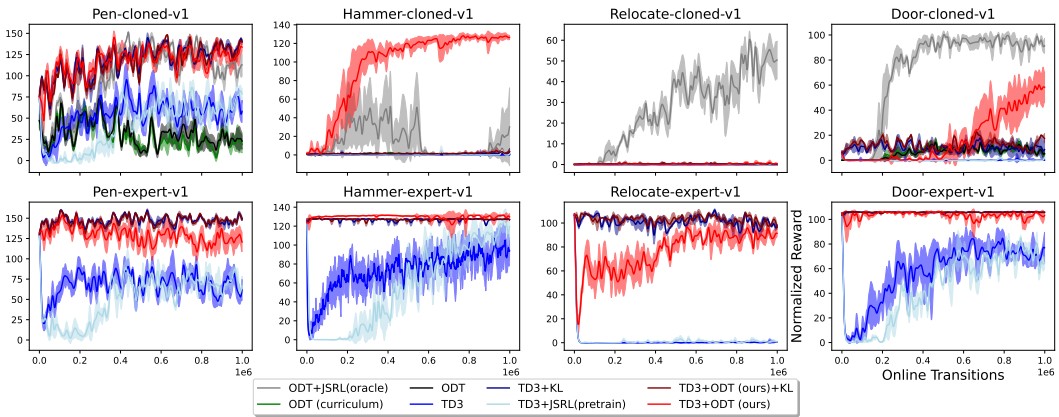

Figure 24: The result of ODT with better exploration (only in cloned dataset) and TD3/TD3+ODT forgetting mitigation. The result shows that 1) ODT with curriculum RTG does not work; 2) even with exploration supported by an oracle, ODT can still fail on some environments such as hammer; 3) JSRL with the pretrained policy does not work for forgetting mitigation; 4) KL regularizer effectively addresses the issue on expert environments for both TD3 and TD3+ODT, but it can hinder the improvement of TD3+ODT with low return.

## G.8  Other possible exploration improvement techniques

In Sec. 3.1, we state that ODT cannot explore the high-RTG region when pretrained with low-quality offline data, and we ran a simple experiment to verify this (Fig. 2). In this section, we test two potential alternatives for addressing the exploration problem: JSRL [59] and curriculum learning.

For JSRL, an expert policy is used for the first $n$ steps in an episode, before ODT takes over. We set $n = 100$ (100% max episode length for adroit pen, 50% max episode length for other adroit environments) initially, and apply an exponential decay rate of $0.99$ for every episode. We test two settings of JSRL: the expert policy being the offline pretrained policy, and the expert policy being *oracle*, i.e., an IQL policy trained on the Adroit expert dataset.

For curriculum learning, we use ODT with a gradually increasing target RTG with the current RTG for rollouts being $\text{RTG}_{\text{eval}} - 0.99^N(\text{RTG}_{\text{eval}} - \text{RTG}_{\text{data}})$. Here, $N$ is the number of episodes sampled in online stage, and $\text{RTG}_{\text{data}}$ is the average RTG of the offline dataset.

Results are summarized in Fig. 24. We found that curriculum RTG does not work, probably because the task is too hard and cannot be improved by random exploration without gradient guidance. Further, even with oracle exploration, ODT is not guaranteed to succeed: it fails on the hammer environment where TD3+ODT succeeds, probably because of insufficient expert-level data and an inability to improve with random exploration.

## G.9  Ablations on the Architecture

In this section, we further examine the source of the performance gain of our method compared to TD3+BC. There are two key differences as stated in Sec. 3.2: The architecture and RL via Supervised (RvS) learning [18]. We can hence assess two more baselines: TD3+BC with our transformer architecture and TD3+RvS using TD3+BC's architecture. We present the ablation result on the Adroit cloned environment in Fig. 25. The result shows that only TD3+BC with our architecture works (albeit still worse than our method). We hypothesize that this is because a simple MLP is hard to model the complicated policy which takes both RTG and state into account.

To further assess if simply adding more layers to the MLP works, we conduct an ablation on the number of layers for TD3+RvS. The result is illustrated in Fig. 26. It shows that simply adding a few layers to the MLP does not aid performance. We speculate that it is probably the transformer architecture that helps modeling the state-and-RTG-conditioned policy.

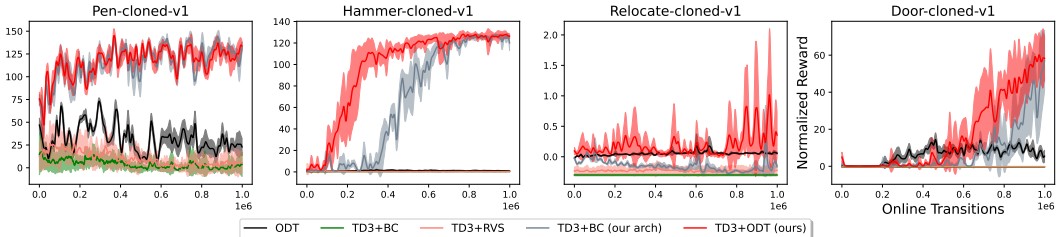

Figure 25: The result of ODT and TD3+BC ablations (TD3+RVS, DDPG+ODT, TD3+BC with our architecture and curriculum RTG for ODT) on Adroit environments. The result shows that only TD3+BC with our architecture works. However, it remains worse than our method.

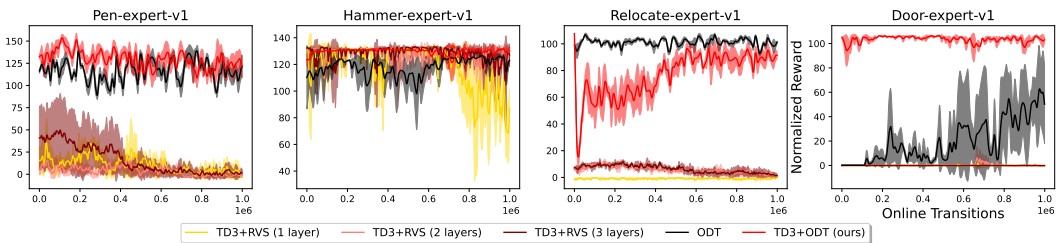

Figure 26: Results of adding more layers to TD3+RvS. The result shows that simply adding MLP layers does not help TD3+RvS match the performance of ODT and TD3+ODT.

## H  Computational Resources

We conduct all experiments with a single NVIDIA RTX 2080Ti GPU on an Ubuntu 18.04 server equipped with 72 Intel Xeon Gold 6254 CPUs @ 3.10GHz. Mujoco experiments takes about $6 - 8$ hours, and the bottleneck is the gradient update; about $50\%$ time is spent on backpropagation and update of parameters. Our critic appended to ODT only takes up about $20\%$ time to train, in which $90\%$ of the critic training time is spent on decision transformer inference to get action for "next state". For the actor, the training overhead of our method is negligible since it only contains an MLP critic inference to get the Q-value. Therefore, overall our method only uses $20\%$ extra time compared to ODT for training, but attains much better results.

## I  Dataset and Algorithm Licenses

Our code is developed upon multiple algorithm repositories and environment testbeds.

**Algorithm Repositories.** We implement our method on the basis of online decision transformer repository, which has a CC BY-NC 4.0 license. We also refer to IQL [28], PDT [64] and TD3+BC [20] repository when running baselines, all of which have MIT licenses.

**Environment Testbeds.** We utilize OpenAI gym [8], MuJoCo [58], and D4RL [19] as testbed, which have an MIT license, an Apache-2.0 license, and an Apache-2.0 license respectively.

