# OpenReview forum: "Reinforcement Learning Gradients as Vitamin for Online Finetuning Decision Transformers"
_NeurIPS.cc/2024/Conference — NeurIPS 2024 spotlight_

### Official Review · Reviewer_NWdD · 2024-07-11

**Soundness:** 2
**Presentation:** 3
**Contribution:** 3
**Rating:** 6
**Confidence:** 4

**Summary:**

The authors consider the problem of online fine-tuning decision transformers. Current approaches for this problem do not work well when the offline data is low-quality. The authors analyze the online DT algorithm theoretically and show why DT-induced policy update would not work if the RTG used for conditioning is much higher than RTG seen in the dataset. They then propose combining the ODT update with RL gradients, in particular, the ones obtained from the TD3 model. This approach was then tested through Adroit, Antmaze, and Mujoco experiments, where the method obtained strong results. Finally, the authors perform additional analyses and make improvements to their proposed approach.

**Strengths:**

- The work considers the important problem of finetuning decision transformers. I see this as a significant issue, as we need strong online fine-tuning methods to better leverage pre-trained models.
- The proposed method seems to achieve quite good performance in the presented experiments, at least when not taking the statistical significance of the results into account. In my opinion, the authors also consider a sufficient set of benchmarks and baselines.
- The study is backed by theoretical results.
- Related work is well-described to the best of my knowledge.
- The appendix includes many additional results and extended analysis that might be of interest to the reader.

**Weaknesses:**

- The empirical results seem to be noisy, and a more careful approach to statistical significance is needed. Looking at, e.g., Figure 5, I'm not sure if TD3+ODT is better than TD3 as the plots overlap quite a lot, but the corresponding Table 1 makes it look like TD3+ODT is a clear winner.
    - This includes bolding "the best solution" in some of the tables. If one of the solutions achieves a score of 101 and the other of 100, in the RL regime, it is usually not possible to tell which one is statistically better and the fair thing to do is to bold both.
    - I didn't find the information on how many seeds are used for each experiment, and this is very important in the RL context.
    - In summary, I would suggest the authors adopt the recommended practices of [1] to have a clearer picture of the results.
- I think the baselines considered here should be made stronger
    - The TD3 baseline collapses right after fine-tuning starts in many cases. It looks like it might be caused by forgetting which was shown to be a problem in RL fine-tuning [2]. I wonder if combining TD3 with some forgetting mitigation technique would help.
    - The authors point out that their method improves on TD3+BC in two ways: architecture (MLP -> DT) and objective (BC -> RvS). I think these two axes of improvement should be investigated separately. That is, introducing two additional baselines would show us where the improvements come from: TD3+BC+Transformer and TD3+RvS (with MLP rather than DT).

[1] Agarwal, Rishabh, et al. "Deep reinforcement learning at the edge of the statistical precipice." Advances in neural information processing systems 34 (2021): 29304-29320. \
[2] Wolczyk, Maciej, et al. "Fine-tuning Reinforcement Learning Models is Secretly a Forgetting Mitigation Problem." Forty-first International Conference on Machine Learning.

**Questions:**

Currently, I would be inclined to accept the paper if not for the problems with the evaluation. As such, I would like to ask the authors:
- Please discuss and address the questions about the evaluation setting, as raised in the Weaknesses section above. If this is sufficiently resolved, I'd be happy to increase the score.
- Please consider introducing stronger baselines. This is not as crucial, but it would make the paper stronger.

Less important but interesting to discuss:
- The authors say: "training a transformer-based value function estimator is quite hard." I agree with this opinion based on my own experiences, but I don't think there has been a thorough discussion of this in the literature. The papers referenced by the authors do not really address this issue well in my opinion. So, it's not a weakness of the paper, but I would like to ask the authors where they think the problems of value learning in transformers come from (or maybe even their online training in general).

**Limitations:**

Although the authors do discuss the limitations of their work, I think there are at least two more points that should be discussed:
* The method has not been tested on image-based datasets and environments.
* There is an increased computational cost as compared to a regular ODT, since we have to train the value functions.

---

> ### Author Rebuttal · Authors · 2024-08-06
>
> Thanks for appreciating our work. Here are our responses:
>
> **Q1. Results are noisy, and a more careful approach to statistical significance is needed.**
>
> Thanks for pointing this out! We understand the importance of reliable evaluation and statistical significance. For this we reported standard deviations in our learning curves in all main results (Fig. 2,3,5) and most ablations (Fig. 6-7, 9, 16-17, 19). We provided Tab. 1 so that readers can focus on the final performance and its gain over pretraining.
>
> To make our reported results more reliable, according to the reviewer’s advice, we evaluated all main results (MuJoCo, adroit, maze, antmaze) using the rliable library, as suggested by the paper mentioned in the review. Results are provided in Fig. 3 of the global pdf, which shows: our method indeed outperforms all baselines on adroit, MuJoCo, and umaze/medium antmaze. Also, we will update Tab. 1-3, 10-11 by highlighting scores >=90% of the best reward with bold-face following MAHALO [1].
>
> **Q2. Number of random seeds.**
>
> Thanks for pointing this out. We use 3 seeds for all experiments (a few ablations without standard deviation use only 1 seed). Note, even without ablations this amounts to a large number of experiments: 1) we use 46 different datasets for our main results [4 for maze, 24 for MuJoCo ({random, medium-replay, medium}x{hopper, halfcheetah, walker2d, ant}x{normal, delayed reward}), 12 for adroit ({pen, hammer, door, relocate}x{expert, cloned, human}), and 6 for antmaze]; and 2) we test 6 methods. This amounts to 800+ runs (46 datasets, 6 methods, 3 seeds), each of which requires 6-8h on average (IQL and TD3+BC need less time, but PDT and other methods on complex environments such as adroit require more).
>
> **Q3. Try adding forgetting mitigation methods onto TD3 baseline.**
>
> Great suggestion! The paper mentioned by the reviewer is pretty recent. It discusses 4 types of forgetting mitigation methods: Behavioral Cloning (BC), Kick Starting (KS), Episodic Memory (EM), and Elastic Weight Consolidation (EWC). Our TD3 baseline and the MLP TD3+BC baseline already use EM as a mitigation method (quote, “simply keeping the examples from the pre-trained task in the replay buffer when training on the new task”).
>
> Note, since TD3 is deterministic without log-likelihood, adapting EWC is out of scope because it requires the Fisher information matrix, which in turn needs the gradient of the log-likelihood. This is also true for BC and KS, which are auxiliary losses of KL-divergences between policies on states drawn from different samplers. However, since BC/KS are essentially soft constraints that prevent the current policy from being too different from the pretrained policy, we can implement BC/KS on continuous action space for deterministic policies by substituting KL divergence with Euclidean distance between actions predicted by the current and pretrained policies. Here, we add a $0.05\cdot\|\|a_{\text{new}}-a_{\text{old}}\|\|_2^2$ penalty term to the actor loss, which is a gradual shift from BC (sampled from offline dataset) to KS (sampled from online rollouts).
>
> To see whether other forgetting mitigation methods work, we also implemented jump-start RL [2]. Concretely, the pretrained policy is used for the first $n$ steps in an episode, before ODT takes over. We set $n=100$ (100% max episode length for pen, 50% max episode length for other adroit environments) initially, and apply an exponential decay rate of $0.99$ for every episode.
>
> We evaluate both methods on adroit environments with the expert dataset (where the forgetting issue is serious) and the cloned dataset (where the agent needs to improve from low-return policies). Results are summarized in Fig. 1. We find that 1) jump-start RL struggles (possibly because it does not directly prevent out-of-distribution updates), and 2) while BC/KS effectively improves both TD3 and TD3+ODT on expert datasets, it also hinders online policy improvement on most cloned datasets.
>
> **Q4. Try other baselines such as TD3+BC+transformer and TD3+RvS.**
>
> We would like to point out that our work is focused on analyzing and improving ODT. Thus the main baseline for this work is ODT instead of TD3+BC. That being said, we agree that TD3+BC+transformer and TD3+RvS are interesting baselines to test. The results are shown in Fig. 2 in the global pdf. We test the methods on adroit environments with the cloned dataset. The results show that TD3+RvS slightly improves upon TD3+BC, while TD3+BC+transformer improves more. We speculate that an MLP is not expressive enough to model the policy change over different RTGs. Note, both suggestions still fall short of our method.
>
> **Q5. Where does the problem of value learning come from?**
>
> Good question! Generally, we found that there is a tradeoff between more information and training stability: while longer context length allows the decision transformer to utilize more information, it also causes increased input complexity, i.e., noise from the environment leads to reduced stability and slower convergence in training. Note, this differs from LLM works, where context length improves generalization thanks to extremely high expressivity, huge amounts of data, and less noisy evaluation. This is especially the case with bootstrapping involved, as the fitting target depends on its own output. In fact, Parisotto et al. [4] suggest that if a recurrent agent can first learn a Markovian policy and value function, then the following training could be stabilized.
>
> **Q6. Some limitations are left out, e.g., lack of image-based testbed and increased computational cost.**
>
> Thanks for pointing these out! Those are important questions, and we will discuss those in a revised limitation section. Regarding computational cost, we discussed the impact of the RL gradient on training in Appendix H: we found that our method is only marginally (20%) slower than ODT. For the MuJoCo experiment, ODT needs ~6 hours to train on our machine.

---

> > ### Comment · Reviewer_NWdD · 2024-08-08
> >
> > I appreciate the thorough response from the authors. After reading it as well as the other reviews, I decided to increase my score.
> >
> > Here's my detailed response:
> > * I appreciate that the authors ran the rliable library to get more stable results. As for the results in Fig. 3 -- are these averaged over all the settings (e.g., cloned and expert in Adroit case)? It would be nice to also see this analysis on each setting.
> > * I understand that having more seeds is expensive, but I think we need them to be sure about the results. Given some overlaps in Figure 3 presented by the authors I would recommend adding at least 2 more seeds (5 in total)
> > * I really appreciate the forgetting mitigation experiments. It makes sense that BC works well on expert datasets but not on cloned ones. I wonder if one could apply some sort of selective forgetting mitigation on the cloned datasets to remember what is useful and forget all of the behaviors that are not useful.
> > * It's quite interesting that the transformer architecture seems to be much more important than the conditioning (i.e., BC vs RvS). I wonder if the transformer is crucial or just having bigger MLPs or RNNs/SSMs would suffice.

---

> > > ### Author Response · Authors · 2024-08-09
> > >
> > > Thanks a lot for your timely reply and for appreciating our rebuttal. Below are our follow-up responses:
> > >
> > > **Q1. About the results in Fig. 3.**
> > >
> > > Yes, those results are averaged over all the settings due to the 1-page limit of the response pdf. For example, for adroit, we average over 12 different settings ({expert, cloned, human}x{pen, hammer, relocate, door}). We will update figures on each setting in the revised paper. To begin, here, we provide some numbers for IQM (which is more robust than median as suggested by the paper mentioned in the review) and optimality gap on the adroit pen environment, as the mean and standard deviation are already provided in the main paper. The numbers in parentheses are the 95% confidence interval.
> > >
> > > **IQM**
> > >
> > > Method | pen-expert-v1 | pen-cloned-v1 | pen-human-v1
> > > ---|---|---|---|
> > > TD3+BC | 40.15 (24.00~48.17)  | -2.39 (-2.82~-1.55) | 18.63 (8.49~34.17)
> > > IQL | **149.44 (146.54~151.13)** | 76.06 (72.17~78.46) | 98.13 (91.18~101.85)
> > > ODT | 129.31 (125.10~133.52) | 26.24 (19.74~32.73) | 32.28 (25.23~39.33)
> > > PDT | 27.91 (14.50~41.32) | 14.05 (5.14~30.09) | 2.67 (1.75~3.58)
> > > TD3 | 69.48 (61.52~77.44) | 70.52 (67.25~73.79) | 32.28 (19.25~45.31)
> > > TD3+ODT (ours) | 118.30 (102.80~135.14) | **130.51 (124.43~135.65)** | **116.71 (110.05~120.61)**
> > >
> > > **Optimality Gap**
> > >
> > > Method | pen-expert-v1 | pen-cloned-v1 | pen-human-v1
> > > ---|---|---|---|
> > > TD3+BC | 59.85 (48.17~76.00) | 102.39 (101.55~102.83) | 81.36 (65.83~91.51)
> > > IQL | **0.00 (0.00~0.00)** | 23.94 (21.54~27.83) | 2.93 (0.00~8.81)
> > > ODT | **0.00 (0.00~0.00)** | 73.75 (67.26~80.25) | 67.61 (60.66~74.76)
> > > PDT | 72.08 (58.68~85.50) | 85.95 (69.90~94.85) | 97.33 (96.42~98.24)
> > > TD3 | 30.51 (22.56~38.48) | 29.48 (26.21~32.75) | 67.72 (54.69~80.74)
> > > TD3+ODT (ours) | **0.00 (0.00~0.00)** | **0.00 (0.00~0.00)** | **0.00 (0.00~0.00)**
> > >
> > > **Q2. More seeds are needed for experiments.**
> > >
> > > We completely agree, thorough evaluation is important. We started experiments with two additional seeds, and we will include those in the revised paper.
> > >
> > > **Q3. Selective forgetting mitigation on cloned datasets.**
> > >
> > > 1. “Forgetting all the behaviors that are not useful” might be suboptimal. While expert trajectories are certainly more useful, it is hard to say which behaviors are not useful at all in the offline dataset. For example, all trajectories may provide information on environment dynamics and possibly undesirable states. We speculate that it would be better if such data could be memorized in some way not reflected in the evaluation policy, e.g., dynamic models in model-based RL, areas of state-action space with low Q-values, or encoded in the policy but “hidden” by conditioning on high RTG. Our method is a combination of the latter two.
> > >
> > > 2. We think the ODT gradient in TD3+ODT during the online phase could be interpreted as a kind of selective forgetting mitigation. Since TD3+ODT (as well as our TD3 / original ODT) keeps the offline data in the replay buffer for ODT itself during the online phase, the supervised learning gradient on those offline data can be regarded as a regularization towards the RTG-conditioned empirical policy, which, ideally, should be equal to the pretrained policy if the algorithm converges during the offline phase.
> > >
> > > 3. Given the performance increase due to the KL regularizer used with TD3+ODT on adroit expert datasets, we agree that an RTG-weighted regularizer could be interesting to test and could potentially further improve results. We have started some experiments, and will add results on this in the revised paper.
> > >
> > > **Q4. Is the transformer crucial, or bigger MLPs would suffice?**
> > >
> > > Good question! While ODT is the main baseline of our paper as we mentioned before, we feel ablations on using bigger MLPs / RNNs for TD3+RvS are interesting. We started some experiments for our revised paper.

---

### Official Review · Reviewer_gVzs · 2024-07-12

**Soundness:** 2
**Presentation:** 3
**Contribution:** 3
**Rating:** 6
**Confidence:** 3

**Summary:**

This paper presents a method for fine-tuning decision transformers (DT) online. The proposal is to integrate TD3 with DT such that the gradients from TD3 can help the online policy to explore highly rewarding trajectories, hence further improving the agent performance. It can be seen as a variant of TD3+BC with the MLP being replaced by a transformer. The experimental results show the proposed method outperforms several baselines, and is effective even when the offline dataset is of low quality, a setting which traditional online DT cannot do well.

**Strengths:**

- The topic of improving DT for online fine-tuning is worth exploring. The way this paper approaches the topic is reasonable.
- The empirical results are good.

**Weaknesses:**

- The TD3 gradients for policy update are not clear. For example, in equation 2, it seems that $Q$ is a constant if you optimize $\mu^{RL}$? Or is $a_t$ predicted by $\mu^{RL}$?
- Some notations are a bit confusing. For example, there are RTG, RTG_real, RTG_eval. Better to clearly define them first.

**Questions:**

- Could you please explain more about the point (2) in line 182: how does DT prioritize trajectories with higher return?

**Limitations:**

- Compared to TD3+BC, the proposed method uses transformer instead of MLP. It costs much more computation to gain better performance. Such limitation is not mentioned in the paper.

---

> ### Author Rebuttal · Authors · 2024-08-06
>
> Thanks for appreciating our work. Below are responses to questions:
>
> **Q1. TD3 gradient for policy update is not clear.**
>
> Thanks for pointing this out! There is a typo in Eq. (2) which should read as follows:
>
> $$\min\_{\mu^{\text{DT}}}\mathbb{E}\_{\tau\sim D}\Bigg[\frac{1}{T\_{\text{train}}}\sum\_{t=1}^{T\_{\text{train}}}\left[-\alpha Q\_{\phi\_1}(s\_t,\mu^{\text{DT}}(s\_{0:t},a\_{0:t-1},\text{RTG}\_{0:t},\text{RTG}=\text{RTG}\_{\text{real}})) + \|\|\mu^{\text{DT}}(s\_{0:t},a\_{0:t-1},\text{RTG}\_{0:t},\text{RTG}=\text{RTG}\_{\text{real}})-a\_t\|\|\_2^2\right]\Bigg],$$
>
> i.e., the $a_t$ in Q should be the action generated by the decision transformer.
>
>
>
> **Q2. Some notations are confusing.**
>
> We introduced RTG in line 82-83 (“RTG, the target total return”), RTG_eval in line 88 (“a desired return RTG_eval is specified”) and line 96, and RTG_real in line 100 (the real return-to-go). We understand that they are not introduced concurrently, which makes it a bit confusing. We will modify this in the revised version.
>
> **Q3. How does DT prioritize trajectories with higher return as mentioned in point (2) in line 182?**
>
> DT prioritizes trajectories because the actions are learned to be generated conditioned on the Return-To-Go (RTG), and we use high RTG as the condition in evaluation. Consider as an example training with two sets of trajectories extending from the same state: one set with ground truth RTG being 0 and the other set with ground truth RTG being 1. If inference asks DT to generate trajectories conditioned on RTG=1, then the generated trajectory will be more similar to the latter set of training trajectories.
>
> **Q4. The limitation that the use of transformers requires more computational cost is not mentioned.**
>
> Thanks for pointing this out. We are happy to include a discussion in the limitation section in the revised version. Meanwhile, we would like to mention two points:
>
> 1. Our work aims to analyze and improve ODT. We thus compare computational cost primarily to ODT instead of MLP-based solutions;
>
> 2. In Appendix H we stated that our proposed solution is only marginally (20%) slower than ODT: while the use of RL gradients slows down training, the actor training overhead is negligible (since it only contains an MLP critic inference to get the Q-value), and the critic only takes about 20% time to train. We found that on our machine, for the MuJoCo experiment, ODT requires about 6 hours to train.

---

> > ### Comment · Reviewer_gVzs · 2024-08-10
> >
> > Thanks for your rebuttal. Given the idea is interesting and the experimental results are thorough, I will adjust my score to 6.

---

> > > ### Author Response · Authors · 2024-08-10
> > >
> > > Thanks a lot for appreciating our work and rebuttal! We will revise our paper as discussed in the reviews for the next version.

---

### Official Review · Reviewer_1gFB · 2024-07-12

**Soundness:** 3
**Presentation:** 3
**Contribution:** 3
**Rating:** 7
**Confidence:** 3

**Summary:**

The authors introduce a novel framework for improving the performance of Online Decision Transformer through adding TD3 gradients to the fine-tuning ODT objective. This is motivated by a theoretical analysis of ODT, that highlights an issue with low-reward, sub-optimal pretraining. The authors also provide an extensive empirical investigation.

**Strengths:**

* Paper is clearly written, and well motivated. The detailed explanation on the preliminaries was useful to the reader.
* The contribution of TD3+ODT appears novel, original, and of high significance to the community.

**Weaknesses:**

* Overclaim of the contribution of “We propose a simple yet effective method to boost the performance of online finetuning of decision transformers.” The empirical results of relocate-expert-v1 show the opposite, where ODT outperforms TD3+ODT. Perhaps the authors can discuss this result and/or refine the claim to environments with particular properties where the expectation is that TD3+ODT outperforms ODT.
* Minor: Section 3.1 could highlight the conceptual figure of the plot of Figure 1, c, to help the reader understand the graph at the beginning of section 3.1.

**Questions:**

* Can the authors comment or perform an ablation with ODT+DDPG for all the main experiments as well?
* What is the impact of context length on your approach?
* For the main experiments how many random seed runs are each experimental result over? E.g. in Figure 2.

**Limitations:**

Limitations are adequately discussed in section 6.

---

> ### Author Rebuttal · Authors · 2024-08-06
>
> Thanks for appreciating our work. Below are responses to questions:
>
> **Q1. Overclaim of “boosting performance”, and the need to refine claims with properties where TD3+ODT should outperform ODT.**
>
> Thanks for pointing this out. While TD3+ODT is generally better than ODT, we are happy to rephrase our claim to reflect that there are some cases where ODT performs better than TD3+ODT. To be specific, we expect ODT to struggle (either fail completely or fail to further improve) with medium-to-low quality offline data. This is supported by our result on adroit environments with cloned/human dataset, antmaze, and MuJoCo (especially medium-replay and random where there are many trajectories with low ground truth RTGs). In contrast, if offline data has good quality (e.g., adroit expert), or the reward signal is sparse (e.g., delayed reward in Appendix G.1) so that RL struggles, then we expect ODT to work as good or better.
>
> **Q2. [minor] Figure in Section 3.1 could be improved for better understanding.**
>
> Thanks a lot for the advice! We will modify the figure accordingly in the revised version.
>
> **Q3. Ablations on DDPG+ODT.**
>
> Great suggestion! Unfortunately, due to the rebuttal time limit, we can only provide ODT+DDPG results on some environments for now. We will provide results of ODT+DDPG on all additional environments in the revised version. The current results are shown in Fig. 2 of the global pdf. We find DDPG+ODT does not work. We speculate that this is due to the rougher landscape of the estimated Q-value without smoothing, delayed update, and double Q-learning by TD3.
>
> **Q4. Impact of context length on the approach.**
>
> We conducted ablations on both the context length at training time ($T_{\text{train}}$) and evaluation / rollout time ($T_{\text{eval}}$). The former is shown in Fig. 18 of Appendix G.3, and the latter is shown in Fig. 4 (b). Generally, we found that there is a tradeoff between more information and training stability: while longer context length allows the decision transformer to utilize more information, it also causes increased input complexity, i.e., noise from the environment leads to reduced stability and slower convergence in training. Note, this differs from LLM works, where context length improves generalization thanks to extremely high expressivity, huge amounts of data, and less noisy evaluation. This is especially the case when bootstrapping is involved, as the fitting target depends on its own output. In fact, in prior work, Parisotto et al. [4] suggest in their Sec. 3.1 that if a recurrent agent can first learn a Markovian policy and value function at the start of its training, then the training process could be stabilized.
>
>
> **Q5. Number of random seeds.**
>
> We use 3 seeds for all experiments (a few ablations without standard deviation use only 1 seed). Note, even without ablations this amounts to a large number of experiments: 1) we use 46 different datasets for our main results [4 for maze, 24 for mujoco ({random, medium-replay, medium}x{hopper, halfcheetah, walker2d, ant}x{normal, delayed reward}), 12 for adroit ({pen, hammer, door, relocate}x{expert, cloned, human}), and 6 for antmaze]; and 2) we test 6 methods. This amounts to 800+ runs (46 datasets, 6 methods, 3 seeds), each of which requires 6-8h on average (IQL and TD3+BC need less time, but PDT and other methods on complex environments such as adroit require more).

---

> > ### Comment · Reviewer_1gFB · 2024-08-11
> >
> > I thank the authors for their detailed rebuttal response, and I particularly appreciate the new experimental ablations in the global response. Thank you for re-phrasing your claim. I look forward to seeing the results of ODT+DDPG on all additional environments in the revised version. I am still keeping my score the same.

---

> > > ### Author Response · Authors · 2024-08-11
> > >
> > > Thanks for your appreciation of our work and response! We have started the experiments and will surely have those results ready in our revised version.

---

### Official Review · Reviewer_AdzZ · 2024-07-13

**Soundness:** 3
**Presentation:** 4
**Contribution:** 3
**Rating:** 7
**Confidence:** 4

**Summary:**

This paper addresses the challenge of online finetuning of online decision transformers (ODT). Theoretical results are provided to show the target return-to-go can hamper finetune. The authors propose to have TD3 gradients added to ODT finetune and improve its performance especially when pretrained with low-reward data. Extensive empirical results are provided to show the proposed method can achieve stronger or competitive results across a large number of environments.

**Strengths:**

**originality**
- The idea to combine TD3 gradient and ODT training is quite novel. The theoretical results can also be considered novel findings.

**quality**
- Overall good quality, the paper is well-written. Figures are easy to read and extensive experiments and ablations are provided.

**clarity**
- Overall clear and easy to follow.

**significance**
- The empirical results help improve our understanding of DT methods and how to better finetune them. The results are quite strong. The extensive experiments across different benchmarks make the results more convincing. The theoretical results add to significance.

**Weaknesses:**

- Maybe I missed something but I feel it might be good to have ODT baselines that try to tackle online finetuning with alternative (and more naive) methods. For example, authors argue that a main difficulty when finetuning DT is that when pretrained with low-reward data, the target RTG at finetuning stage is simply hard to obtain. But if we provide a realistic target RTG initially (based on pretrain data), and gradually increase it, will that help performance? Another thing is if we have a better exploration policy, will that help ODT finetune to the same extent?
- Adding the TD3 component can make the training more complex and slower.

**Questions:**

- Will alternative strategies that improve exploration at finetune phase help ODT to the same extent?
- The random datasets are not commonly tested in previous papers, when the baselines are tested on these datasets, are the hyperparameters tuned for these datasets?

**Limitations:**

Some limitations are discussed in section 6.

---

> ### Author Rebuttal · Authors · 2024-08-06
>
> Thanks for appreciating our work. Below are responses to questions:
>
> **Q1. test ODT baselines that tackle online finetuning with alternative methods, e.g., better exploration policy and gradually increasing target RTG.**
>
> Great suggestions! Currently, ODT explores due to the stochasticity of its policy. We enhance ODT in two ways and provide results in the global pdf: 1) ODT+gradually increasing target RTG (which is similar to curriculum learning in RL); and 2) a **cheating** baseline ODT with **oracle exploration**, i.e., jump-start RL [2], using as the guide policy an expert policy trained via IQL on expert data. Concretely, the expert policy is used for the first $n$ steps in an episode, before ODT takes over. We set $n=100$ (100% max episode length for adroit pen, 50% max episode length for other adroit environments) initially, and apply an exponential decay rate of $0.99$ for every episode.
>
> Results are summarized in Fig. 1 of the global pdf. Curriculum RTG does not work, probably because the task is too hard and cannot be improved by random exploration without gradient guidance. Also, even with oracle exploration, ODT is not guaranteed to succeed: it fails on the hammer environment where TD3+ODT succeeds, probably because of insufficient expert-level data and an inability to improve with random exploration.
>
> **Q2. TD3 slows down training.**
>
> **Our method is only 20% slower than ODT without TD3 gradients.** In Appendix H, we discussed the impact of the introduced RL gradient on training: the critic only takes about 20% of the time to train, while the actor overhead is negligible (since it only contains an MLP critic inference to get the Q-value). We found that on our machine, for the MuJoCo experiment, ODT requires about 6 hours to train. We will add part of the discussion in Appendix H to the limitation section of the revised paper.
>
> **Q3. Are the hyperparameters tuned on the random datasets?**
>
> No, as suggested in Tab. 9 in the appendix, we use the same hyperparameters for every variant (e.g., medium / medium-replay / random, expert / human / cloned) on the same environment. Note, our parameters on the random datasets are aligned with the ODT medium environment and not tuned further.

---

### Author Rebuttal · Authors · 2024-08-06

We thank the reviewers, ACs and SACs for valuable feedback. We are delighted that our idea was appreciated as novel (AdzZ, 1gFB), well-motivated (1gFB), valuable (NWdD, gVzs, 1gFB) and backed by theoretical foundations (AdzZ, NWdD), the literature review was appreciated as well-described (NWdD), and results were referred to as strong (AdzZ, gVzs) and sufficiently extensive (AdzZ, NWdD). Also, reviewers unanimously rate the presentation of our work positively (AdzZ gives “excellent”, all other reviewers give “good”).

We answer common questions here:

**Q1. Number of random seeds (1gFB, NWdD).** We use 3 seeds for all experiments (a few ablations without standard deviation use only 1 seed). Note, even without ablations this amounts to a large number of experiments: 1) we use 46 different datasets for our main results [4 for maze, 24 for mujoco ({random, medium-replay, medium}x{hopper, halfcheetah, walker2d, ant}x{normal, delayed reward}), 12 for adroit ({pen, hammer, door, relocate}x{expert, cloned, human}), and 6 for antmaze]; and 2) we test 6 methods. This amounts to 800+ runs (46 datasets, 6 methods, 3 seeds), each of which requires 6-8h on average (IQL and TD3+BC need less time, but PDT and other methods on complex environments such as adroit require more).

**Q2. Our design slows down training (AdzZ, gVzs, NWdD).** We thank the reviewers for pointing this out, and we are happy to include the discussion in the limitation section. Meanwhile, we note two points:

1. Our work aims to analyze and improve ODT. We thus compare computational cost primarily to ODT instead of MLP-based solutions;

2. In Appendix H we stated that our proposed solution is only marginally (20%) slower than ODT: while the use of RL gradients slows down training, the actor training overhead is negligible (since it only contains an MLP critic inference to get the Q-value), and the critic only takes about 20% time to train. We found that on our machine, for the MuJoCo experiment, ODT requires about 6 hours to run.

**Q3. Additional ablations (AdzZ, 1gFB, NWdD).** We also provide a variety of new experimental ablations, which are summarized in the global pdf (experiment 1 and 5 mentioned below are in Fig. 1; experiment 2, 3, 6 are in Fig. 2; experiment 4 is in Fig. 3).

**1. Better exploration by “cheating”: letting an expert take over in early steps (reviewer AdzZ).** Even with an oracle (“cheating”) exploration strategy, ODT can still fail in some cases where our method prevails, likely because of insufficient expert-level data and an inability to improve with random exploration.

**2. ODT baselines which gradually increase target RTG (reviewer AdzZ).** We find that such curriculum learning does not work, probably because the task is too hard and cannot be improved by random exploration without gradient guidance.

**3. ODT+DDPG (reviewer 1gFB).** We find that ODT+DDPG does not work. We speculate that this is due to a less stable Q-value landscape during training.

**4. Better evaluation using rliable library (reviewer NWdD).** We re-evaluate all our main results using the rliable library, and find that our method still generally outperforms other baselines, especially in adroit environments.

**5. TD3 with forgetting mitigation issues implemented via KL regularizer and jump-start RL where the guide policy is the pretrained policy (reviewer NWdD).** Jump-start RL with the pretrained policy does not work well, probably because it does not directly prevent out-of-distribution policy updates. KL regularizer effectively mitigates forgetting for both TD3 and TD3+ODT, but it also hinders policy improvement of the TD3+ODT policy pretrained on low-RTG offline data.

**6. TD3+BC+transformer and TD3+RvS (reviewer NWdD).** We find that TD3+BC+transformer works well in the adroit environment, albeit still worse than our proposed method. TD3+RvS does not work: it only slightly outperforms TD3+BC. We speculate that this is because an MLP is not expressive enough to model the policy change over different RTGs.

Finally, we list the papers referred to in our responses:

**References**

[1] A. Li et al. MAHALO: Unifying Offline Reinforcement Learning and Imitation Learning from Observations. In ICML, 2023.

[2] I. Uchendu et al. Jump-Start Reinforcement Learning. In ICML, 2023.

[3] S. Emmons et al. RVS: What is Essential for Offline RL via Supervised Learning? In ICLR, 2022.

[4] E. Parisotto et al. Stabilizing Transformers for Reinforcement Learning. In ICML, 2020.

---

### Public Comment · ~Pranav_Poduval1 · 2026-05-07
**Does TD3+ODT actually address the same support/coverage problem identified for ODT, or only add a heuristic local regularizer?**

Great work. I have been interested in Decision Transformers as an alternative to offline RL, and I may be misunderstanding some parts since I have only read original Decision Transformers & Online Decision Transformers papers, so I would appreciate clarification or any misunderstandings due to lack o background in this domain

The paper’s motivation is that ODT can fail when the requested RTG is outside the return support of the offline/online replay data. I agree this is a plausible failure mode. However, I am not convinced that adding TD3 gradients theoretically addresses this issue, rather than adding a useful local actor-critic regularizer.

The paper argues that high-target-RTG conditioning depends on quantities such as

$$
p_\beta(G=g\mid s,a),\quad p_\beta(G=g\mid s),
$$

which are poorly estimated when (g) is outside data support. But the proposed TD3 actor term introduces

$$
\nabla_a Q_\phi(s,a)\big|*{a=\mu*\theta(z)},
$$

which is also reliable only in state-action regions covered by the replay buffer or subsequent online exploration. If the critic is queried at unsupported actions, then (Q_\phi(s,a)), and especially its action gradient, are function-approximation extrapolations. This seems like the standard offline/off-policy critic extrapolation problem.

Could the authors clarify what assumption makes the TD3 gradient reliable precisely where ODT lacks coverage? For example, is there any bound such as

$$
|\nabla_a Q_\phi(s,a)-\nabla_a Q^\pi(s,a)|\le \epsilon
$$

on the actor’s queried state-action distribution, or any support/concentrability condition such as

$$
d^{\pi_\theta}(s,a)\le C d_{\mathcal B}(s,a)?
$$

Without such a condition, it seems the TD3 term may inherit a similar support problem rather than solve it.

A related question is whether the method is theoretically doing more than TD3+BC with a DT actor. The actor loss is essentially

$$
|\mu_\theta(z_t)-a_t|^2
-----------------------

\lambda Q_\phi(s_t,\mu_\theta(z_t)).
$$

Locally, for small (\lambda), this suggests

$$
\mu_\theta(z_t)
\approx
a_t+
\frac{\lambda}{2}\nabla_a Q_\phi(s_t,a_t),
$$

i.e., a behavior-cloning action plus a critic-gradient nudge. This is reasonable, but it looks like a TD3+BC-style regularized actor-critic objective, with the main difference being that the actor is a Decision Transformer. What is the DT-specific theoretical contribution beyond this architectural substitution?

I also found the AWAC/Laplace derivation stronger than justified. The paper assumes both

$$
p_\beta(G\mid s,a)=\mathrm{Laplace}(Q^\beta(s,a),\sigma)
$$

and

$$
p_\beta(G\mid s)=\mathrm{Laplace}(V^\beta(s),\sigma).
$$

But generally,

$$
p_\beta(G\mid s)=\int \beta(a\mid s)p_\beta(G\mid s,a),da.
$$

Thus, if action-conditioned return distributions are Laplace with different centers, the marginal is generally a mixture of Laplace distributions, not a single Laplace centered at (V^\beta(s)). Is this lemma intended only as intuition, or does any main claim rely on it?

Finally, could the reported gains be explained by benign local coverage in the selected benchmarks? The method may work when the replay buffer already contains enough local state-action variation for TD3 to learn useful gradients, or when online exploration quickly expands coverage. A useful diagnostic would be an environment where the offline data contains poor actions with no local gradient information toward the better action, or where the optimal action lies in an unsupported action region. Does TD3+ODT still improve there?

Overall, I think the empirical heuristic may be useful, but the theoretical framing appears overstated. The paper convincingly argues that ODT can fail due to return-coverage issues, but I do not see a corresponding argument that TD3 gradients overcome those issues rather than adding a local actor-critic update valid only under usual replay-support assumptions.

---

> ### Public Comment · ~Kai_Yan1 · 2026-05-08
>
> Hi Pranav,
>
> Thank you for your interest in our work and for providing valuable feedback. Here are our responses to your concerns:
>
> **Q1. The paper’s motivation and the choice of TD3.**
>
> 1. The observation that "ODT can fail when the requested RTG is outside the return support of the offline/online replay data" constitutes only one aspect of our motivation. A complementary motivation: gradients from the value function during online RL provide crucial first-order information on the current policy to improve the RTG—a signal that ODT natively struggles to utilize (relying at most on zeroth-order information).
>
> 2. TD3 is designed for **online improvement.** Because it continually interacts with the environment and updates the buffer with data from the current policy, it inherently circumvents the strict "queried at unsupported action" problem during its online application. It is **not meant for better addressing Q-value estimation on unsupported actions in a purely offline setting.** We acknowledge that many specialized offline RL methods do this better than TD3, such as IQL (https://arxiv.org/abs/2110.06169), Cal-QL (https://arxiv.org/pdf/2303.05479), MCQ (https://arxiv.org/abs/2206.04745), and ACA (https://github.com/ZishunYu/Actor-Critic-Alignment), many of which explicitly handle lack of data coverage by mitigating possible spikes in the Q-value landscape.
>
> While we pretrain TD3 offline for some environments ($\alpha>0$ in Eq. (2)), this is primarily to align the offline and online phases so that the critic generates reasonable initial gradients. We agree that integrating more advanced offline RL gradients could yield further performance gains; we have investigated IQL in our paper (Appendix C) and leave further exploration with state-of-the-art offline RL gradients as a direction for future work.
>
> To sum up, our work does not aim to provide theoretical Q-value bounds or support/concentrability guarantees. Instead, our focus is to analyze the failure mode of RTG-conditioned self-supervised finetuning and to show that adding local RL gradients can help to address this failure mode.
>
> **Q2. Regarding Lemma 2.**
>
> We agree that an action-conditioned return distribution being Laplace does not imply that its marginal return distribution is also Laplace. A mixture of Laplace distributions need not be Laplace. Lemma 2 should therefore be read as relying on a simplifying assumption on the relevant RTG distributions, rather than as deriving the marginal distribution from the conditional one.
>
> As we explicitly stated in the paper in Appendix E.2 near the end of page 20, the Laplace distribution assumption is strict and impractical for real-life applications. The purpose of the lemma is to provide intuition for why DT-style updates can resemble advantage-weighted updates under a simplified distributional model. Our core failure analysis does not mathematically rely on this conditional-to-marginal implication.
>
> **Q3. Regarding explanation on the reported gains.**
>
> Our single-state, one-step, one-dimensional continuous-action MDP in Figs. 1 and 2 is designed as a controlled example of the scenario you described, where the optimal action lies in a region not covered by the offline data. In this example, the reward peak is hidden from the offline dataset, ODT fails to discover it through RTG-conditioned supervised finetuning, while DDPG/ODT+DDPG can use critic gradients to identify a local direction of improvement.
>
> For high-dimensional continuous-control environments, we do not claim that every performance gain comes from exactly the same unsupported-action mechanism as in the synthetic MDP. In particular, MuJoCo datasets may still contain broad action coverage. Rather, these experiments support the broader mechanism studied in the paper: when pretrained from low-return data, ODT often struggles to translate high RTG conditioning into effective online policy improvement, while RL gradients provide a useful local improvement signal. This is consistent with the substantial gains we observe over ODT on MuJoCo datasets, as well as the strong final performance of our method on challenging Adroit environments such as hammer-cloned-v1, door-cloned-v1, and door-human-v1.
>
> Thank you again for carefully reading our paper. We hope these responses answer your questions.

---

> > ### Public Comment · ~Pranav_Poduval2 · 2026-05-15
> >
> > Thanks a lot for the detailed response, it does help!

---

### Decision · Program_Chairs · 2024-09-25

**Decision:**

Accept (spotlight)

**Comment:**

This paper has been unanimously recommended for acceptance. It tackles an interesting problem—the offline-to-online problem in reinforcement learning—applied to the decision transformer and provides some theoretical and empirical results. In one way or another, the rebuttal addressed most of the authors' concerns, and I'm recommending the acceptance of this paper.